# Characterizing the risk of fairwashing

**Ulrich Aïvodji**
École de Technologie Supérieure *
`ulrich.aivodji@etsmtl.ca`

**Hiromi Arai**
RIKEN Center for Advanced Intelligence Project
`hiromi.arai@riken.jp`

**Sébastien Gambs**
Université du Québec à Montréal
`gambs.sebastien@uqam.ca`

**Satoshi Hara**
Osaka University
`satohara@ar.sanken.osaka-u.ac.jp`

## Abstract

Fairwashing refers to the risk that an unfair black-box model can be explained by a fairer model through post-hoc explanation manipulation. In this paper, we investigate the capability of fairwashing attacks by analyzing their fidelity-unfairness trade-offs. In particular, we show that fairwashed explanation models can generalize beyond the suing group (*i.e.,* data points that are being explained), meaning that a fairwashed explainer can be used to rationalize subsequent unfair decisions of a black-box model. We also demonstrate that fairwashing attacks can transfer across black-box models, meaning that other black-box models can perform fairwashing without explicitly using their predictions. This generalization and transferability of fairwashing attacks imply that their detection will be difficult in practice. Finally, we propose an approach to quantify the risk of fairwashing, which is based on the computation of the range of the unfairness of high-fidelity explainers.

## 1 Introduction

As machine learning models are increasingly integrated into the pipeline of high-stakes decision processes, concerns about their transparency are becoming prominent and difficult to ignore for the actors deploying them. As a result, post-hoc explanation techniques have recently gained popularity as they may appear as a potentially viable solution to regain trust in machine learning models' predictions. More precisely, post-hoc explanation techniques refer to methods used to explain how black-box ML models produce their outcomes [26, 9]. Current existing techniques for post-hoc explanations include global and local explanations. In a nutshell, global explanations focus on explaining the whole logic of the black-box model by training a surrogate model that is interpretable by design (*e.g.*, linear models, rule-based models or decision trees) while maximizing its fidelity to the black-box model. In contrast, local explanations aim at explaining a single decision by approximating the black-box model in the vicinity of the input point through an interpretable model.

However, a growing body of works has recently shown that post-hoc explanation techniques not only can be misleading [38, 46] but are also vulnerable to adversarial manipulations, wherein an adversary misleads users' trust by devising deceiving explanations. This phenomenon has been demonstrated for a broad range of post-hoc explanation techniques, including global and local explanations [4, 49], example-based explanations [25], visualization-based explanations [31, 20] and counterfactual explanations [36]. For instance, in a fairwashing attack [4], the adversary manipulates the explanations to under-report the unfairness of the black-box models being explained. This attack can significantly impact individuals who have received a negative outcome following the model's prediction while depriving them of the possibility of contesting it.

---

*Work done while at Université du Québec à Montréal

35th Conference on Neural Information Processing Systems (NeurIPS 2021).

The fundamental question regarding fairwashing attacks is their manipulability, which we define as the ability to maximize the fidelity of an explanation model under an unfairness constraint. The manipulability of fairwashing attacks directly impacts the possibility to detect them. Indeed, if the manipulability is so low that the explanation manipulation can be detected, the risk of fairwashing is small and misleading decisions can be avoided. By contrast, if the manipulability is high enough the manipulation is undetectable, we will be under threat of the use of unfair models whose unfairness is hidden by malicious model producers through manipulated explanations. In this work, in the context of fairwashing for global explanations, we provide the first empirical results demonstrating that the manipulability of fairwashing is likely to be high. To assess the manipulability of fairwashing, we used the fidelity-unfairness trade-off and evaluated two characteristics of fairwashing, namely generalization and transferability.

- **Generalization of fairwashing beyond the suing group.** In a fairwashing attack, a manipulated explanation is tailored specifically for a suing group of interest so that the explanation is fair within this group. As the explanation is specific to that group, we hypothesize that the same explanation can fail for another group. Based on this hypothesis, we assess the manipulability of fairwashing through its generalization capability. Our results suggest that the fidelity of the fairwashed explanation evaluated on another group is comparable to the one evaluated on the suing group. This means that the above hypothesis is negative, in the sense that explanations built to fairwash a suing group can generalize to another group not explicitly targeted by the attack.

- **Transferability of fairwashing beyond the targeted model.** In the fairwashing attack, a manipulated explanation is targeted specifically for the deployed black-box model. However, in practical machine learning, it is usually the case that the deployed model is updated frequently. Thus, we hypothesize that there can be an inconsistency between the manipulated explanations provided to the suing group in the past and the currently deployed model. Based on this hypothesis, we quantify the manipulability of fairwashing through its transferability. Our results suggest that the fidelity of the fairwashed explanation evaluated on another model is comparable to the one evaluated on the deployed black-box model. Thus, the above hypothesis is also negative as fairwashed explanations designed for a specific model can also transfer to another model.

**Implications to undetectability.** We observed the generalization and transferability of fairwashing attacks on several datasets, black-box models, explanation models and fairness criteria. As a consequence, our results indicate that detecting manipulated explanations based on the change of fidelity alone is not a viable solution (or at least it is very difficult).

**Another way of quantifying fairwashing manipulability.** In the above experiments, the manipulability of fairwashing was evaluated using the fidelity of explanation and its changes. Our negative results suggest that fidelity alone may not be an effective metric for quantifying the manipulability of fairwashing. Thus, we further investigated a different way of quantifying the manipulability of fairwashing using the `Fairness In The Rashomon Set` (FaiRS) [17] framework. Our results indicate that this framework can be effectively used to quantify the manipulability of fairwashing.

**Related work.** In the context of example-based explanations' manipulation, Fukuchi et al. [25] have demonstrated the risk of stealthily biased sampling, which occurs when a model producer explains the behaviour of its black-box model by sampling a subset of its training dataset. Slack et al. [49] have shown that variants of LIME [45] and SHAP [39], two popular post-hoc local explanation techniques, can be manipulated to underestimate the unfairness of black-box models. Following the same line of work, Le Merrer and Trédan [37] have demonstrated that a malicious model producer can always craft a fake local explanation to hide the use of discriminatory features. Another work by Laugel et al. [36] has focused explicitly on the use of counterfactual explanations, which is a form of example-based explanation. They have demonstrated that generated instances can be unjustified (*i.e.*, not supported by ground-truth data) for most techniques existing for such type of explanation. Visualization-based explanation techniques can also be manipulated, as shown by recent works on saliency map-based explanations' manipulation. Heo et al. [31] have demonstrated that these types of explanations are vulnerable to the so-called adversarial model manipulation while Dombrowski et al. [20] have shown their vulnerability to adversarial input manipulation. In our previous work [4], we introduced the notion of fairwashing as a rationalization exercise. We devised `LaundryML`, an algorithm that can systematically rationalize black-box models' decisions through global or local explanations. The details of this method, which form the basis of the fairwashing attacks in this study,

are given in Section 2.2. A subsequent work [35] has also investigated the possibility that black-box models can be explained with high fidelity by global interpretable models whose features are very different from that of the black-box and look innocuous. Finally, several negative aspects of post-hoc explanations have been reported in recent studies. For instance, post-hoc explanations can mislead model designers when they are used for debugging purposes [1, 2] or can be leveraged to perform powerful model stealing attacks [42, 6].

**Outline.** In Section 2, we review the preliminary notions necessary to understand this work, and we describe how to explore the fidelity-unfairness trade-offs of a fairwashing attack by using an $\epsilon$-constraint method to solve its underlying multi-objective optimization problem. In Section 3, we present the results obtained from our study for a diverse set of datasets, black-box models, explanation models and fairness metrics. In Section 4, we consider quantifying the manipulability of fairwashing using the `Fairness In The Rashomon Set` (FaiRS) [17] framework. Finally, we discuss the main implications of our findings in Section 5 as well as its limitations and societal impact in Section 6.

## 2 Setting and problem formulation

### 2.1 Notations

Let $X \in \mathcal{X} \subset \mathbb{R}^n$ denote a feature vector, $Y \in \mathcal{Y} = \{0, 1\}$ its associated binary label (for simplicity we assume a binary classification setup without loss of generality) and $G \in \mathcal{G} = \{0, 1\}$ a feature defining a group membership (*e.g.*, with respect to a sensitive attribute) for every data point sampled from $\mathcal{X}$. In addition, we assume that $b : \mathcal{X} \to \mathcal{Y}$ refers to a black-box classifier of a particular model class $\mathcal{B}$ (*e.g.*, neural network or ensemble model) mapping any input $X \in \mathcal{X}$ to its associated prediction $\hat{Y} \in \mathcal{Y}$. Finally, let $e : \mathcal{X} \to \mathcal{Y}$ be a global explanation model from a particular model class $\mathcal{E}$ (*e.g.*, linear model, rule list or decision tree) designed to explain $b$. In the context of fairwashing attacks, which we formalize in Definition 2, we refer to the data instances on which the attack is performed as the *suing group*.

In this work, we measure the unfairness $\text{unf}_{\mathcal{D}}(f)$ of a model $f$ on a dataset $\mathcal{D}$ by using *statistical notions of fairness* [13, 15, 16, 30], which require a model to exhibit approximate parity according to a statistical measure across the different groups defined by the group membership $G$. In particular, we consider four different statistical notions of fairness, namely statistical parity [22, 13, 33, 23, 51], predictive equality [15, 16], equal opportunity [30] and equalized odds [15, 34, 30, 50]. The definitions of these fairness metrics are listed in Table 1.

Table 1: Summary of the different statistical notions of fairness considered.

| Fairness notion | Definition |
|---|---|
| Statistical Parity ($\Delta_{\text{SP}}$) | $\lvert P(\hat{Y} = 1 \mid G = 0) - P(\hat{Y} = 1 \mid G = 1) \rvert$ |
| Predictive Equality ($\Delta_{\text{PE}}$) | $\lvert P(\hat{Y} = 1 \mid Y = 0, G = 0) - P(\hat{Y} = 1 \mid Y = 0, G = 1) \rvert$ |
| Equal Opportunity ($\Delta_{\text{EO}_{\text{pp}}}$) | $\lvert P(\hat{Y} = 1 \mid Y = 1, G = 0) - P(\hat{Y} = 1 \mid Y = 1, G = 1) \rvert$ |
| Equalized Odds ($\Delta_{\text{EOdds}}$) | $\lvert P(\hat{Y} = 1 \mid Y = 1, G = 0) - P(\hat{Y} = 1 \mid Y = 1, G = 1) \rvert$ and $\lvert P(\hat{Y} = 1 \mid Y = 0, G = 0) - P(\hat{Y} = 1 \mid Y = 0, G = 1) \rvert$ |

### 2.2 Problem formulation

Our investigation is motivated by our previous work [4] in which we defined fairwashing in global and local explanations as a manipulation exercise in which high-fidelity and fairer explanations can be designed to explain unfair black-box models.

**Definition 1 (Global explanation fidelity)** *Let $b$ be a black-box model, $e$ a global explanation model for $b$, and $X$ a set of data instances. Following the definition in [18], the fidelity of $e$ with respect to $b$ on $X$ is expressed as:*

$$\text{fidelity}(e) = \frac{1}{\lvert X \rvert} \sum_{x \in X} \mathbb{I}(e(x) = b(x)).$$

**Definition 2 (Global fairwashing attack)** *Let $b$ be a black-box model and $X_{sg}$ a set of data instances hereafter referred to as suing group. A global fairwashing attack consists in finding an interpretable global model $e = p(b, X_{sg})$ derived from the black-box $b$ and the suing group $X_{sg}$ using some attack process $p(\cdot, \cdot)$, such that $e$ is fairer than $b$ for a given fairness metric.*

To realize this, in our previous work [4], we devised `LaundryML`, an algorithm that can systematically fairwash unfair black-box models' decisions through both global and local explanations. `LaundryML` is a constrained model enumeration technique [29, 28] that searches for explanation models maximizing the fidelity while minimizing the unfairness for a given unfair black-box model.

In this study, we go a step further by determining the fidelity-unfairness trade-offs of the fairwashing attack and characterizing the *manipulability* of the fairwashed explanations. For this purpose, we compute the set of Pareto optimal explanation models describing all the achievable fidelity-unfairness trade-offs by solving the following problem:

$$\text{minimize } \mathbb{E}_{(x_i, b(x_i)) \sim \mathcal{D}_{sg}}[l(e(x_i), b(x_i))], \quad \text{subject to } \text{unf}_{\mathcal{D}_{sg}}(e) \leq \epsilon, \tag{1}$$

in which $e$ is the explanation model, $l(e(x), b(x))$ is the loss function (*e.g.*, cross entropy), $\mathcal{D}_{sg} = \{X_{sg}, b(X_{sg})\}$ is formed by the suing group and the prediction of the black-box model $b$ on the suing group, and $\epsilon$ is the value of the unfairness constraint.

## 3 Experimental evaluation

In this section, we evaluate the manipulability of the fairwashing attack over several datasets, black-box models, explanation models, and fairness metrics. We start by replicating the results of our previous study [4], which demonstrate that explanations can be fairwashed. Then, we evaluate two fundamental properties of fairwashing, namely generalization and transferability[2].

**Datasets.** We have investigated four real-world datasets commonly used in the fairness literature, namely Adult Income, Marketing, COMPAS, and Default Credit. Table 2 summarizes the main characteristics of these datasets.

- **Adult Income.** The UCI Adult Income [21] dataset contains demographic information about 48,842 individuals from the 1994 U.S. census. The associated classification task consists in predicting whether a particular individual earns more than 50,000$ per year. We used `gender` (`Female`, `Male`) as group membership for investigating fairness.
- **Bank Marketing.** The UCI Bank Marketing [43] dataset contains information about 41,175 customers of a Portuguese banking institution contacted as part of a marketing campaign (*i.e.*, phone calls), whose goal was to convince them to subscribe to a term deposit. The classification task consists of predicting who will subscribe to a term deposit. We used `age` (`30-60`, `not30-60`) as group membership.
- **COMPAS.** The COMPAS [8] dataset gathers 6,150 records from criminal offenders in Florida during 2013 and 2014. Here, the classification task consists in inferring who will re-offend within two years. We used `race` (`African-American`, `Caucasian`) as group membership.
- **Default Credit.** The Default Credit [21] dataset is composed of information of 29,986 Taiwanese credit card users. The classification task is to predict whether a user will default in its payments. We used `sex` (`Female`, `Male`) as group membership.

**Preprocessing.** Before the experiments, each dataset is split into three subsets, namely the *training set* (67%), the *suing group* (16.5%) and the *test set* (16.5%). We created 10 different samplings of the three subsets using different random seeds and averaged the results of over these 10 samples. The training set is used to learn the black-box models, while the suing group dataset is used to prepare the explanation models as well to evaluate their fidelity-unfairness trade-offs. Finally, the objective of the test set is to estimate the accuracy of the black-box models as well as the generalization of the explanation models beyond their suing groups. For all models (*i.e.,* black-boxes and explanation models), we used a one-hot encoding of the features of the datasets.

---

[2]Our implementations are available at https://github.com/aivodji/characterizing_fairwashing

Table 2: Summary of the datasets used. $N$, $n_f$, $n_{ohe}$ and $n_r$ denote respectively the numbers of data points, the number of features, the number of one-hot encoded features and the number of rules.

| Dataset | Category | $N$ | $n_f$ | $n_{ohe}$ | $n_r$ |
|---|---|---|---|---|---|
| Adult Income | Finance | 48,842 | 11 | 40 | 177 |
| Marketing | Commerce | 41,175 | 20 | 61 | 178 |
| COMPAS | Justice | 6,150 | 8 | 11 | 121 |
| Default Credit | Finance | 29,986 | 23 | 35 | 188 |

**Black-box models.** We have trained four different types of black-box models on each dataset, namely an AdaBoost classifier [24], a Deep Neural Network (DNN), a Random Forest (RF) [12] and a XgBoost classifier [14]. To tune the hyperparameters of these models, during their training, we performed a hyperparameter search with 25 iterations using `HyperOpt` [10]. The performances (*i.e.*, accuracy and unfairness) of the four black-box models on the suing group are provided in Table 3.

Table 3: The performances of the black-box models evaluated on the suing group. Each cell is of the form $\begin{bmatrix} \text{accuracy} & \Delta_{\text{SP}} & \Delta_{\text{PE}} \\ & \Delta_{\text{EOpp}} & \Delta_{\text{EOdds}} \end{bmatrix}$.

|  | AdaBoost | | DNN | | RF | | XgBoost | |
|---|---|---|---|---|---|---|---|---|
| Adult Incomde | 0.85 | .17 .06
.12 .12 | 0.85 | .16 .06
.06 .07 | 0.86 | .16 .06
.09 .09 | 0.86 | .17 .06
.09 .09 |
| Marketing | 0.91 | .10 .04
.13 .13 | 0.91 | .09 .04
.09 .09 | 0.91 | .10 .04
.04 .06 | 0.91 | .10 .04
.08 .09 |
| COMPAS | 0.68 | .24 .25
.14 .25 | 0.68 | .28 .31
.18 .31 | 0.67 | .26 .28
.16 .28 | 0.68 | .27 .30
.18 .30 |
| Default Credit | 0.80 | .03 .04
.01 .04 | 0.81 | .03 .04
.01 .04 | 0.81 | .03 .03
.01 .03 | 0.81 | .02 .02
.01 .03 |

**Fairwashed explanation models.** We solved the optimization problem defined in Equation 1 for three model classes, namely logistic regression, rule lists and decision trees. For rule lists, we used FairCORELS [5], a modified version of CORELS [7], which trains rule lists under fairness constraints. For both logistic regression and decision trees, we used the exponentiated gradient technique [3], which is a model agnostic technique to train any classifier under fairness constraints[3].

### 3.1 Experiment 1: Replicating the result of Aïvodji et al. [4]

We first demonstrate that the explanations can be fairwashed by replicating the results of our prior study [4] over a wide range of datasets, black-box models, explanation models and fairness metrics.

**Setup.** Given a suing group $X_{sg}$, for each black-box model $b$ and each fairness metric $m$, the Pareto fronts are obtained by sweeping over 300 values of fairness constraints $\epsilon_m \in [0, 1]$. For each value of $\epsilon_m$, an explanation model $e_{\epsilon_m}$ is trained to satisfy the unfairness constraint $\epsilon_m$ on $X_{sg}$, by solving the problem in Equation 1 for logistic regression, rule list and decision tree. Then, the effective unfairness and fidelity (with respect to $b$) on $X_{sg}$ are obtained. Finally, the set of non-dominated points is computed.

**Results.**[4] Top rows in Figure 1 show the fidelity-unfairness trade-offs of fairwashed logistic regression explainers found for the four black-box models, respectively on Adult Income and COMPAS, for the suing group, using four different fairness metrics. Figure 2 displays the fidelity of the fairwashed logistic regression explainers when they are designed to be at least 50% less unfair than the black-box models they are explaining. Results are shown for all datasets, fairness metrics and three families

---

[3]We used the `Fairlearn` library [11] implementation: https://github.com/fairlearn/fairlearn
[4]See Appendix C for the results of all the datasets and models.

of black-box models (AdaBoost, DNN and RF). Consistently over all these results, we observe that the fairwashed explanation models found for the suing groups were significantly less unfair than the black-box models while maintaining high fidelity (*i.e.*, the fairwashing attacks were effective as shown in Figure 1). More precisely, for any combination of fairness metric $m$ and black-box model $b$, a fairwashed logistic regression displays an unfairness less than $50\%$ of the unfairness of $b$ while maintaining a fidelity greater than $92\%$, $96\%$, $81\%$ and $90\%$ respectively for Adult Income, Marketing, COMPAS and Default Credit(*c.f.*, Figure 2). Furthermore, the small change in percentage of the fidelity of the fairwashed explanation models indicates that fairwashing does not introduce an important loss in fidelity when compared to non-fairwashed explanations models.

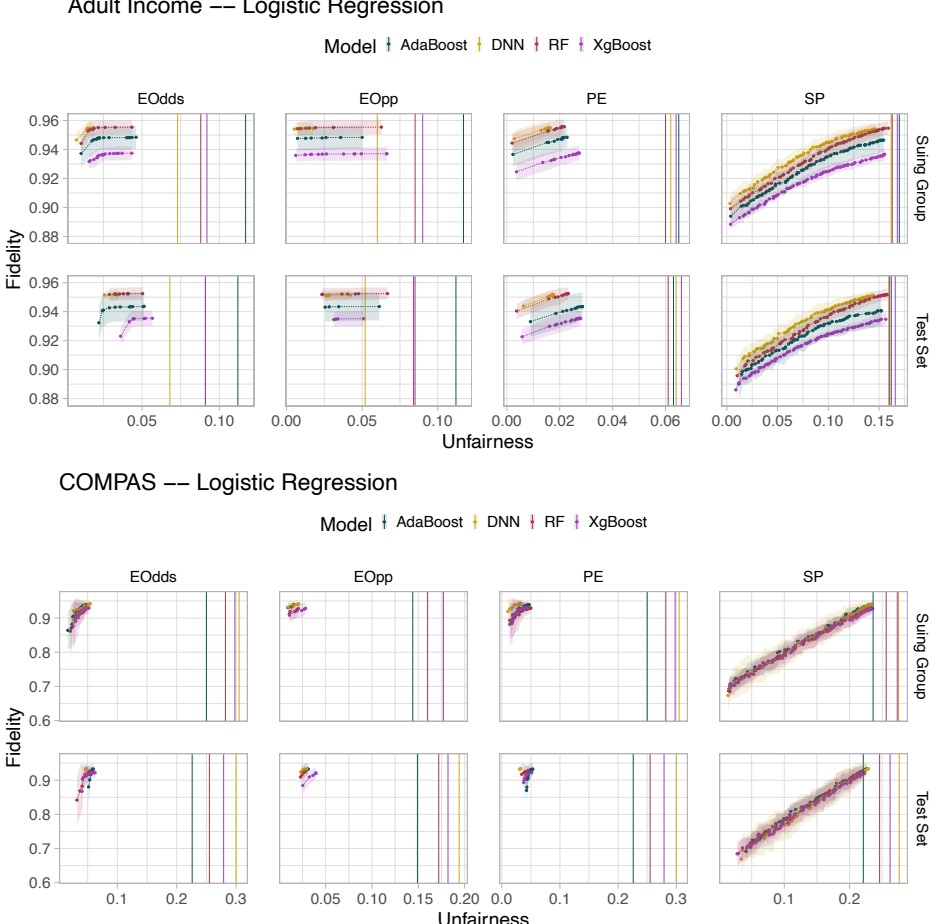

Figure 1: Fidelity-unfairness trade-off of fairwashing attacks for equalized odds ($\Delta_{\mathrm{EOdds}}$), equal opportunity ($\Delta_{\mathrm{EOpp}}$), predictive equality ($\Delta_{\mathrm{PE}}$) and statistical parity ($\Delta_{\mathrm{SP}}$) metrics on Adult Income and COMPAS datasets, using logistic regression as explanation models. Vertical lines denote the unfairness of the black-box models. Results are averaged over 10 fairwashing attacks. The standard deviations are shown as shaded regions.

## 3.2 Experiment 2: Generalization of fairwashing beyond suing groups

We now assess the manipulability of fairwashing attacks in terms of generalization. As a fairwashed explanation is tailored specifically for a suing group of interest, it could be the case that the same explanation can fail for another group. Our results detailed below suggest that this is not the case. That is, the above hypothesis is negative as explanations built to fairwash a suing group can generalize to another group not explicitly targeted by the fairwashing attack.

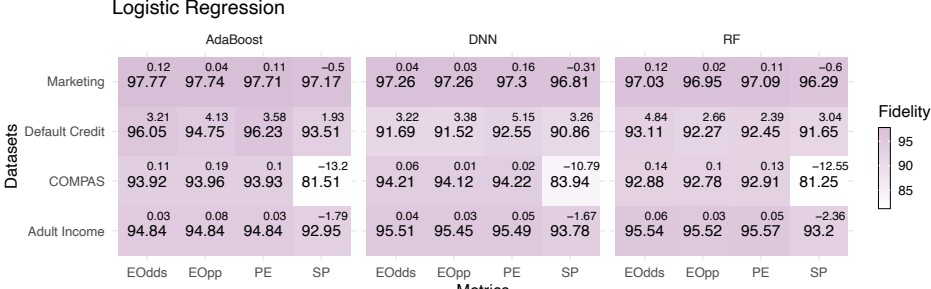

Figure 2: Fidelity of the fairwashed logistic regression explainers that are $50\%$ less unfair than the black-box models they are explaining. Results (averaged over 10 fairwashing attacks) are shown for AdaBoost, DNN and RF black-box models, for all datasets and fairness metrics. The content of each cell is in the form of $x^y$, in which $x$ represents the fidelity of the fairwashed explanation model, and $y$ its percentage change with respect to the fidelity of the unconstrained explainer, used here as a baseline.

**Setup.** We used the same experimental setup as in Experiment 1. However, the unfairness and fidelity of the explanation model are computed on the test set $X_{test}$ such that $X_{sg} \cap X_{test} = \emptyset$. As $X_{test}$ is not disclosed to the model producer, the fairwashed explanation may fail to generalize on $X_{test}$ and exhibit unfairness.

**Results.**[5] Bottom rows in Figure 1 show the fidelity-unfairness trade-offs of fairwashed logistic regression explainers found for the four black-box models on Adult Income and COMPAS, on the test set, using four different fairness metrics. Overall, the results show that the explanation models designed for a particular suing group generalize well also to the test set by achieving similar fidelity-unfairness trade-offs.

**Implications to undetectability.** One might try to detect manipulated explanations by preparing a second suing group unknown to the model producer, with the expectation that the manipulated explanations will fail for that second group. However, the subtle gaps between the trade-offs of the suing group $X_{sg}$ and the test set $X_{test}$ suggest that this would not be a convincing evidence of the occurrence of fairwashing. More precisely, only a fraction of cases exhibiting considerable drops in fidelity could be detected, while the majority of manipulated explanations will go unnoticed.

In addition, the subtle gaps could be made even smaller by adopting robust fairness-enhancing techniques (*e.g.*, [40]). The fairwashing attack defined in Equation 1 is equivalent to a problem of training an explanation model under a fairness constraint, in which the training pair $(X, Y)$ is formed by the suing group $X_{sg}$ and the predictions $b(X_{sg})$ of the black-box $b$. As a result, the issue of generalizing beyond the suing group can be reduced to the problem of generalizing fairness beyond the training set. Thus, robust fairness-enhancing techniques could potentially enable malicious model producers to obtain explanation models with better generalizations and smaller gaps.

### 3.3 Experiment 3: Transferability of fairwashing beyond the targeted model

In this section, we assess the manipulability of the fairwashing attack in terms of transferability. In practical machine learning, it is usually the case that the deployed model is updated frequently. Thus, an inconsistency could occur between the manipulated explanations generated before the model update and the currently deployed model. Our results below suggest that this is not the case, in the sense that the fairwashed explanation for a specific model can transfer to another model.

**Setup.** Given a suing group $X_{sg}$, a teacher black-box model $b_{teacher}$ (corresponding to an old model), a fairness metric $m$, its associated fairness constraint $\epsilon_m$ and a set of student black-box models $b^i_{student}$, with $i = 1, \dots, n$ (corresponding to updated models), an explanation model $e_{\epsilon_m}$

---

[5]See Appendix C for the results of all the datasets and models.

is trained to satisfy the unfairness constraint $\epsilon_m$ on $X_{sg}$, by solving the problem in Equation 1 for logistic regression, rule lists and decision trees. Afterwards, the unfairness and fidelity of $e_{\epsilon_m}$ are evaluated with respect to each of the student black-box models $b^i_{student}$ on $X_{sg}$. For this experiment, we considered four black-box models (AdaBoost, DNN, RF and XgBoost). First, we fixed one model as the teacher model and used the remaining ones as student models. We conducted the experiments for all four possible combinations of the (teacher, student) models. We evaluated the results on a number of unfairness constraints ($\epsilon \in \{0.03, 0.05, 0.1\}$) to simulate both strong and loose fairness constraints.

**Results.**[6]. Figure 3 displays the fidelity and unfairness of fairwashed logistic regression explainers with respect to both the teacher and the set of student black-box models on the suing group $X_{sg}$. Results are shown for Adult Income and COMPAS, for all fairness metrics and different values of the unfairness constraint ($\epsilon = 0.05$). Overall, our results demonstrate that fairwashed explanations can generalize well to student black-box models by displaying high fidelity in many cases. For instance, on Adult Income with a predictive equality constraint set to 0.05, a fairwashed logistic regression explainer that had a fidelity of 95% for a DNN teacher model successfully transferred to AdaBoost, RF and XgBoost student models with a fidelity of respectively 94%, 95% and 93%.

**Implications to undetectability.** If the hypothesis that manipulated explanations are vulnerable to model updates is true, one would expect that the manipulated explanations generated before the model update exhibit an inconsistency with the updated model. However, we observed that high-fidelity explanations can be transferred to another model in many situations. Hence, the change in fidelity after the model update would not be a reliable evidence for detecting fairwashing attacks as it will overlook several manipulated explanations with a small drop of fidelity.

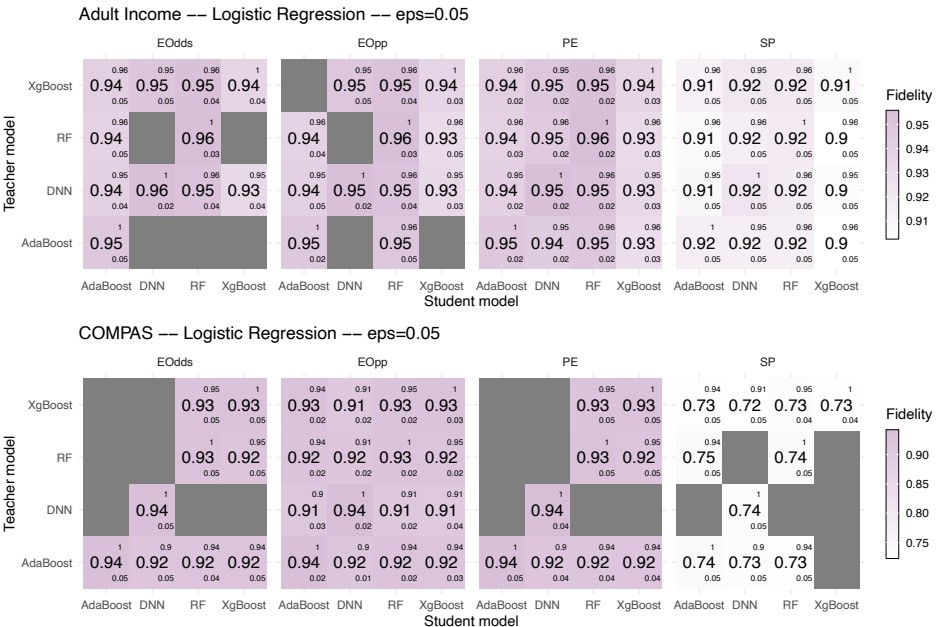

Figure 3: Analysis of the transferability of fairwashing attacks for equalized odds, equal opportunity, predictive parity and statistical parity on Adult Income and COMPAS datasets, for $\epsilon = 0.05$, and for fairwashed logistic regression explainers. The result in each cell is in the form of $x^y_z$, in which $y$ denotes the label agreement between the teacher black-box model and the student black-box model, $x$ is the fidelity of the fairwashed explanation model and $z$ is its unfairness. Blank cells denotes the absence of transferability for the unfairness constraint imposed. Results are averaged over 10 fairwashing attacks.

---

[6]See Appendix D for the results of all the datasets and models.

# 4 Another way of quantifying the manipulability of fairwashing

Our experimental results in the previous section revealed that fairwashed explanations can generalize to unseen suing groups and can transfer to other black-box models with high fidelity. This suggests that fidelity alone may not be an effective metric for quantifying the manipulability of fairwashing. In this section, we consider another way of quantifying the manipulability of fairwashing using the `Fairness In The Rashomon Set` (FaiRS) [17] framework.

At a high level, one can say that fairwashing is possible because the set of all the explanations models for a particular black-box model (at a particular level of fidelity) is diverse in terms of unfairness (*i.e.*, contains both fair and unfair explanation models). This phenomenon has been observed for a broad range of models and domains, and received different names including predictive multiplicity [41], underspecification [19] and multiplicity in the Rashomon set [48, 27, 47].

Given a model class $\mathcal{F}$, a loss function $L_{\mathcal{D}}(\cdot)$ over a dataset $\mathcal{D}$ of interest, a reference model $f^*$ (*e.g.*, optimal explanation model with at least $95\%$ fidelity), and a performance threshold $\tau \in [0, 1]$, the Rashomon set $R_s(\mathcal{F}, f^*, \tau)$ is defined as :

$$R_s(\mathcal{F}, f^*, \tau) = \{f \in \mathcal{F} \mid L_{\mathcal{D}}(f) \leq (1 + \tau)L_{\mathcal{D}}(f^*)\} \tag{2}$$

We can quantify the manipulability of fairwashing by seeking the models with the lowest unfairness within the Rashomon set. More precisely, if the Rashomon set contains explanation models with sufficiently low unfairness, then the risk of fairwashing is high as one can adopt such a model as high-fidelity explanation models with low unfairness. Conversely, if all the models in the Rashomon set have unfairness close to that of the black-box model, the fairwashing will fail as it is not possible to find high-fidelity explanation models with an unfairness significantly lower than that of the black-box model.

A possible way to identify the model with the lowest unfairness in the Rashomon set is to use the method of Coston et al. [17], which computes the range of the unfairness of high-fidelity explanation models by solving the following problem:

$$\text{minimize } \text{unf}_{\mathcal{D}_{sg}}(e), \quad \text{subject to } L_{\mathcal{D}_{sg}}(e) \leq v, \tag{3}$$

in which $e$ is the explanation model, $\mathcal{D}_{sg} = \{X_{sg}, b(X_{sg})\}$ is formed by the suing group and the prediction of the black-box model $b$ on the suing group, while $v$ is the value of the constraint on the loss to explore different levels of fidelity.

**Setup.** In this experiment, we explore different values for the fidelity ranging from $70\%$ to $98\%$. In particular, we solve the problem of minimizing the disparity (and not the absolute disparity) under constraints of the loss [17, Problem 2] using the reduction approach of Agarwal et al. [3] to explore the range of unfairness for a fixed value of the loss. We used this approach and swept over different values of the loss function to compute the range of unfairness of high-fidelity explanation models.[7]

**Results.**[8] Figure 4 shows that logistic regression explainers on Adult Income have a wider range of unfairness in the Rashomon set compared to that of COMPAS. This result implies that the fairwashing attack has a higher manipulability on Adult Income. This observation is coherent with the results observed in Experiments 1 and 2 (*c.f.*, Figures 1 and 2) in which, although fairwashing is possible for both datasets, it is easier (*i.e.*, in terms of the possibility to have high-fidelity with low unfairness) to perform fairwashing on Adult Income than COMPAS.

# 5 Conclusion

In this paper, we have investigated the manipulability of fairwashing attacks by analyzing their fidelity-unfairness trade-offs in diverse situations. In particular, we have demonstrated that fairwashing attacks have high manipulability. Furthermore, we showed that fairwashed explanations can generalize to unseen suing groups and can transfer across black-box models by displaying a high fidelity. The lesson to draw from our investigation is that *relying on the fidelity alone and its changes as proxies*

---

[7]We used the implementation provided by the authors in the public Github repository available at `https://github.com/asheshrambachan/Fairness_In_The_Rashomon_Set`

[8]See Appendix E for the results of all the datasets and black-box models.

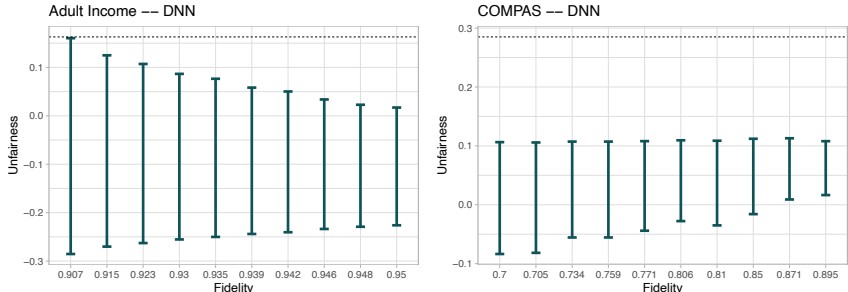

Figure 4: Range of the statistical parity of logistic regression explainers for different values of the fidelity for a DNN black-box trained on Adult Income and COMPAS. Horizontal lines denote the unfairness of the black-box models.

*for the quality of a post-hoc explanation can be misleading* as a fairwashed explanation model can exhibit high fidelity even to unseen suing groups and to other black-box models while being significantly less unfair than the black-box model being explained. Our first result obtained for the generalization of fairwashed explanations beyond suing groups demonstrates that a *fairwashed explanation model can also rationalize subsequent unfair decisions made by the original black-box model for free*. This precludes the possibility of designing fairwashing detection techniques that leverage the instability of the unfairness with respect to variations in the suing group. Indeed, such a technique will most likely fail against fairwashed explanation models designed using stable fair classification algorithms [32, 40]. Our second result, the transferability of fairwashed explanations across black-box models, revealed that model producers can *use fairwashed explanation models to rationalize unfair decisions of future black-box models*. Since fidelity alone did not prove to be an effective measure, we investigated another possible way to quantify the manipulability of fairwashing. To this end, we used the `Fairness In The Rashomon Set` (FaiRS) framework of Coston et al. [17] to compute the range of the unfairness of high-fidelity explainers, and observed that it is possible to quantify the manipulability of fairwashing by using this framework.

**Future work.** In this study, we focused on statistical notions of fairness defined over binary output and binary attributes. These definitions of fairness can be extended to continuous output (*e.g.*, predicted class probability) [44]. However, investigating the manipulability of fairwashing attacks to these extended problems remains open. We also hypothesize that one may be able to design a meta-classifier for detecting fairwashing. Despite the fact that the fidelity of fairwashed explanations is considerably high, it may be possible that some implicit patterns specific to fairwashing could be detected and exploited (*e.g.*, the fidelity can be low on specific subgroups). Thus, if there are several pairs of fairwashed/honest explanations, one might be able to train a meta-classifier to distinguish between fairwashed and non-fairwashed explainers by seeking such patterns.

## 6 Limitations and societal impact

**Limitations.** In this paper, fidelity is defined as the label agreement between the black-box and explanation models. This definition was first introduced in [18] and is still the popular quality measure used in global explanation techniques. While there are several criticisms regarding the use of fidelity as a quality measure, we believe that considering its popularity in the community, it still remains a reasonable choice to raise awareness about the risk of fairwashing.

**Societal impact.** One of the objectives of this paper is to raise awareness about the risks that can occur when post-hoc explanations are used to assess fairness claims. As such, it aims at averting the potential negative societal impacts of such assessments. Nonetheless, it is possible that malicious model producers could use the manipulation attacks presented in this paper to perform fairwashing in the real world. However, in addition to increasing the vigilance of the community, we believe that this paper makes a significant step towards detecting and preventing such fairwashing attacks.

**Acknowledgments**

We thank the anonymous NeurIPS reviewers for their insightful feedback. Hiromi Arai is supported by JST, PRESTO Grant Number JPMJPR1752, Japan. Satoshi Hara is supported by JSPS KAKENHI Grant Number 20K19860, and JST, PRESTO Grant Number JPMJPR20C8, Japan. Sébastien Gambs is supported by the Canada Research Chair program, a Discovery Grant from NSERC, the Legalia project from the AUDACE program funded by the FQRNT, the project *Privacy and Ethics: Understanding the Convergences and Tensions for the Responsible Development of Machine Learning* funded by the Office of the Privacy Commissioner of Canada (OPC) as well as the NSERC-RDC DEEL project. The opinions expressed in this paper are only the one of the authors and do not necessarily reflect those of the OPC.

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
