# A Technical details

**Computing environment.** All the experiments were run on an Intel Core i7 (2.90 GHz with 16GB of RAM). Performing a fairwashing attack for a specific value of the fairness constraint value takes on average 5 minutes to complete on a single CPU.

# B Performances of black-box models

Table 4 summarizes the performances (accuracy, unfairness) of the four types of black-box models (AdaBoost, DNN, RF and XGBoost) on each partition (training set, suing group dataset and testing set) of the four datasets considered.

# C Fidelity-unfairness trade-offs complementary results

Figures 5, 6, 7 and 8 present the fidelity-unfairness trade-off of fairwashing attacks respectively for equalized odds, equal opportunity, predictive equality and statistical parity metrics on Adult Income, COMPAS, Default Credit and Marketing. Results are shown for decision tree, logistic regression and rule list as explanation models.

Figures 9, 10 and 11 present the fidelity of the fairwashed decision tree, logistic regression, and rule list explanation models that are $50\%$ less unfair than the black-box models they are explaining.

# D Complementary results for transferability

Figures 12, 13 and 14 present the performances of the transferability attacks for equalized odds, equal opportunity, predictive parity and statistical parity on Adult Income. Results are shown for different values of the unfairness constraint ($\epsilon \in \{0.03, 0.05, 0.1\}$) and for decision tree, logistic regression and rule list explanation models. Similar results are shown for COMPAS (Figures 15, 16 and 17), Default Credit (Figures 18, 19 and 20) and Marketing (Figures 21, 22 and 23).

# E Complementary results on quantification of the fairwashing risk

Figures 24, 25, 26 and 27 present the range of the statistical parity of logistic regression explanation models of AdaBoost, DNN, RF, and XGBoost black-box models trained respectively on Adult Income, COMPAS, Default Credit and Marketing. Results are shown for different values of the fidelity of the fairwashed explanation models. More precisely, for different values of the unfairness constraint ($\epsilon \in \{0.01k \mid k = 1 \ldots 10\}$), we first trained a fairwashed explanation models $e_\epsilon$. Then, we computed the loss $\tau_\epsilon$ of the $e_\epsilon$. Finally, we used $\tau_\epsilon$ as constraint for the problem defined in Equation 3 to compute the range of the unfairness of all the explanation models that have similar performances as $e_\epsilon$.

Table 4: Summary of the performances (accuracy, unfairness) of the four black-box models (AdaBoost, DNN, Random Forest, and XGBoost) on the train set, test set and suing group of Adult Income, COMPAS, Default Credit and Marketing datasets.

| Dataset | Model | Partition | Accuracy | SP | PE | EOpp | EOdds |
|---|---|---|---|---|---|---|---|
| Adult Income | AdaBoost | Train | 0.86 | 0.17 | 0.06 | 0.11 | 0.11 |
| | | Test | 0.85 | 0.16 | 0.06 | 0.11 | 0.11 |
| | | Suing Group | 0.85 | 0.17 | 0.06 | 0.12 | 0.12 |
| | DNN | Train | 0.86 | 0.16 | 0.06 | 0.05 | 0.06 |
| | | Test | 0.85 | 0.16 | 0.06 | 0.05 | 0.07 |
| | | Suing Group | 0.85 | 0.16 | 0.06 | 0.06 | 0.07 |
| | RF | Train | 0.87 | 0.16 | 0.06 | 0.07 | 0.07 |
| | | Test | 0.85 | 0.16 | 0.06 | 0.08 | 0.09 |
| | | Suing Group | 0.86 | 0.16 | 0.06 | 0.09 | 0.09 |
| | XgBoost | Train | 0.88 | 0.17 | 0.06 | 0.06 | 0.07 |
| | | Test | 0.85 | 0.17 | 0.07 | 0.08 | 0.09 |
| | | Suing Group | 0.86 | 0.17 | 0.06 | 0.09 | 0.09 |
| COMPAS | AdaBoost | Train | 0.68 | 0.22 | 0.23 | 0.15 | 0.23 |
| | | Test | 0.68 | 0.22 | 0.23 | 0.15 | 0.23 |
| | | Suing Group | 0.68 | 0.24 | 0.25 | 0.14 | 0.25 |
| | DNN | Train | 0.68 | 0.27 | 0.28 | 0.18 | 0.28 |
| | | Test | 0.67 | 0.28 | 0.30 | 0.19 | 0.30 |
| | | Suing Group | 0.68 | 0.28 | 0.31 | 0.18 | 0.31 |
| | RF | Train | 0.69 | 0.24 | 0.25 | 0.17 | 0.25 |
| | | Test | 0.67 | 0.25 | 0.26 | 0.17 | 0.26 |
| | | Suing Group | 0.67 | 0.26 | 0.28 | 0.16 | 0.28 |
| | XgBoost | Train | 0.69 | 0.26 | 0.28 | 0.18 | 0.28 |
| | | Test | 0.67 | 0.26 | 0.28 | 0.18 | 0.28 |
| | | Suing Group | 0.68 | 0.27 | 0.30 | 0.18 | 0.30 |
| Default Credit | AdaBoost | Train | 0.81 | 0.03 | 0.05 | 0.02 | 0.05 |
| | | Test | 0.80 | 0.02 | 0.04 | 0.01 | 0.04 |
| | | Suing Group | 0.80 | 0.03 | 0.04 | 0.01 | 0.04 |
| | DNN | Train | 0.82 | 0.03 | 0.04 | 0.01 | 0.04 |
| | | Test | 0.81 | 0.03 | 0.04 | 0.02 | 0.04 |
| | | Suing Group | 0.81 | 0.03 | 0.04 | 0.01 | 0.04 |
| | RF | Train | 0.83 | 0.03 | 0.03 | 0.01 | 0.03 |
| | | Test | 0.81 | 0.02 | 0.02 | 0.01 | 0.03 |
| | | Suing Group | 0.81 | 0.03 | 0.03 | 0.01 | 0.03 |
| | XgBoost | Train | 0.83 | 0.03 | 0.03 | 0.01 | 0.03 |
| | | Test | 0.81 | 0.02 | 0.02 | 0.01 | 0.02 |
| | | Suing Group | 0.81 | 0.02 | 0.02 | 0.01 | 0.03 |
| Marketing | AdaBoost | Train | 0.91 | 0.10 | 0.04 | 0.13 | 0.13 |
| | | Test | 0.91 | 0.10 | 0.04 | 0.17 | 0.17 |
| | | Suing Group | 0.91 | 0.10 | 0.04 | 0.13 | 0.13 |
| | DNN | Train | 0.93 | 0.09 | 0.03 | 0.07 | 0.07 |
| | | Test | 0.91 | 0.09 | 0.04 | 0.07 | 0.08 |
| | | Suing Group | 0.91 | 0.09 | 0.04 | 0.09 | 0.09 |
| | RF | Train | 0.93 | 0.09 | 0.03 | 0.04 | 0.05 |
| | | Test | 0.91 | 0.10 | 0.04 | 0.07 | 0.08 |
| | | Suing Group | 0.91 | 0.10 | 0.04 | 0.04 | 0.06 |
| | XgBoost | Train | 0.92 | 0.10 | 0.04 | 0.08 | 0.09 |
| | | Test | 0.91 | 0.11 | 0.05 | 0.11 | 0.11 |
| | | Suing Group | 0.91 | 0.10 | 0.04 | 0.08 | 0.09 |

## Adult Income –– Decision Tree

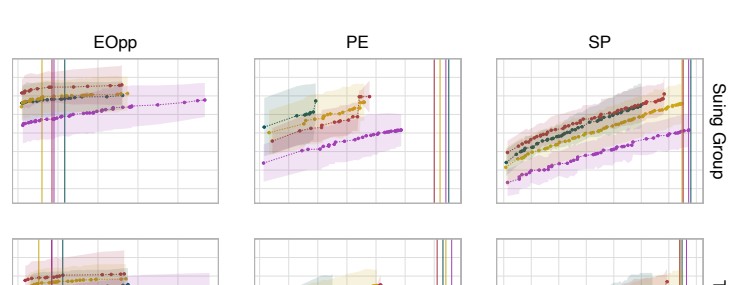

## Adult Income –– Logistic Regression

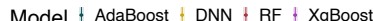
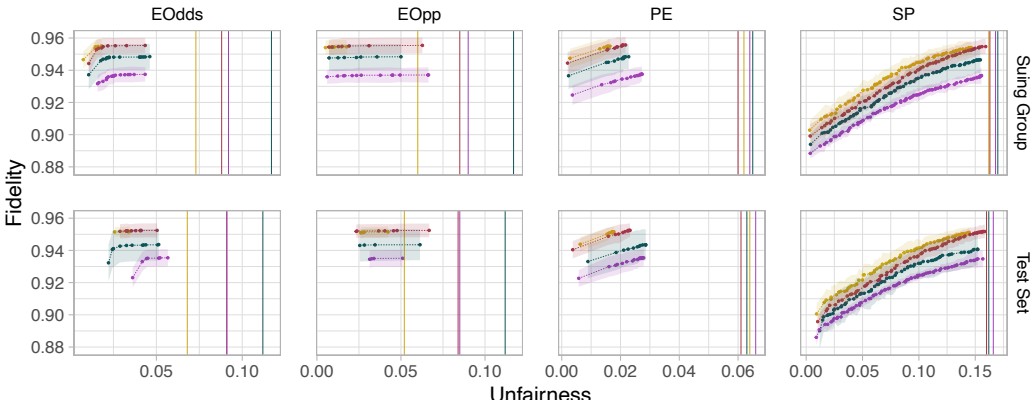

## Adult Income –– Rule List

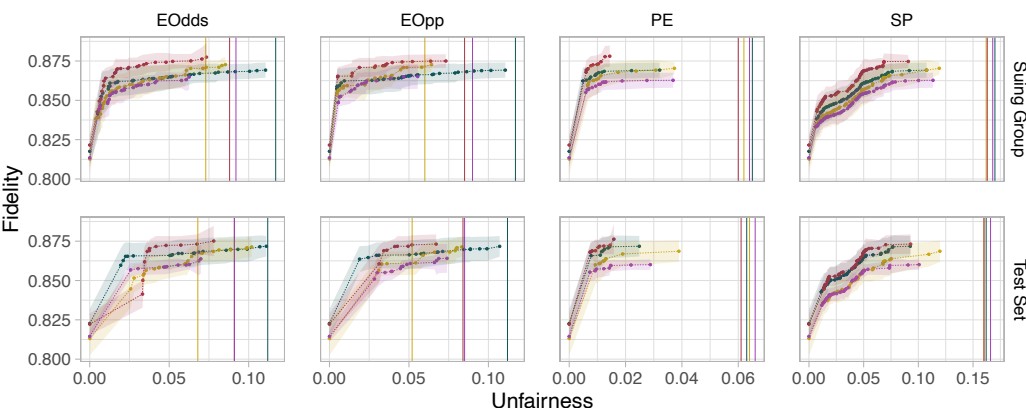

Figure 5: Fidelity-Unfairness trade-off of fairwashing attacks for equalized odds, equal opportunity, predictive equality and statistical parity metrics on Adult Income, using decision tree, logistic regression and rule list as explanation models. Vertical lines denote the unfairness of the black-box models. Results are averaged over 10 fairwashing attacks.

## COMPAS —— Decision Tree

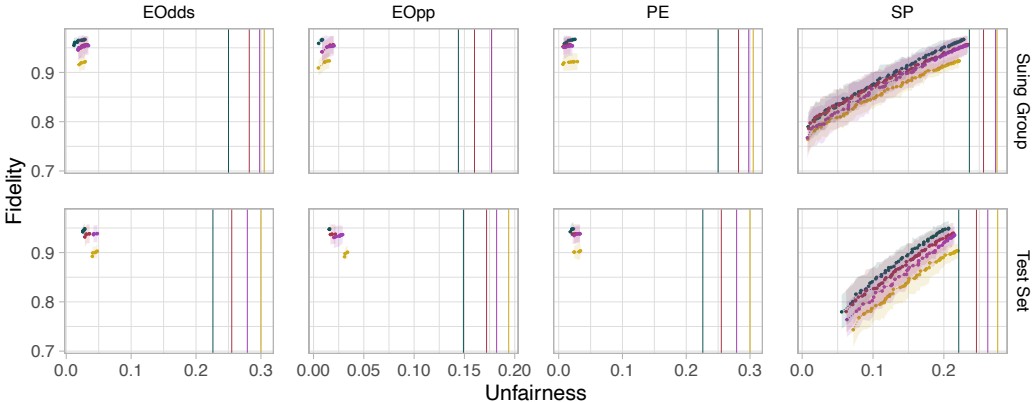

## COMPAS —— Logistic Regression

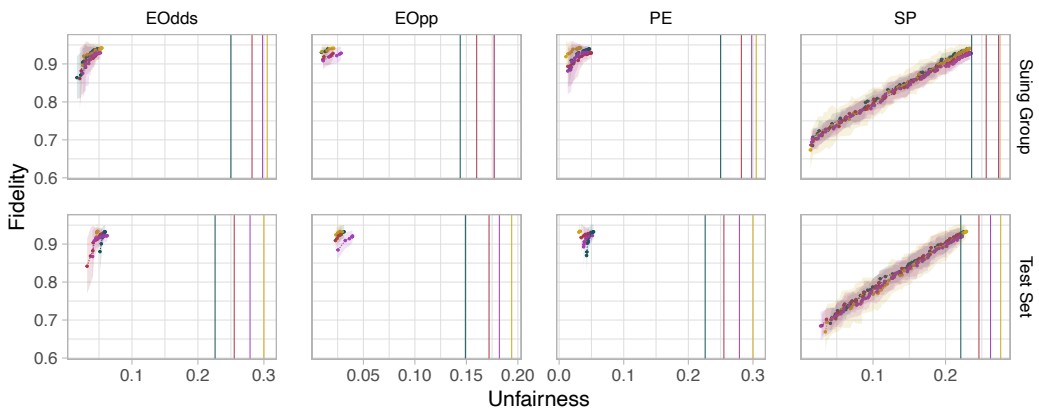

## COMPAS —— Rule List

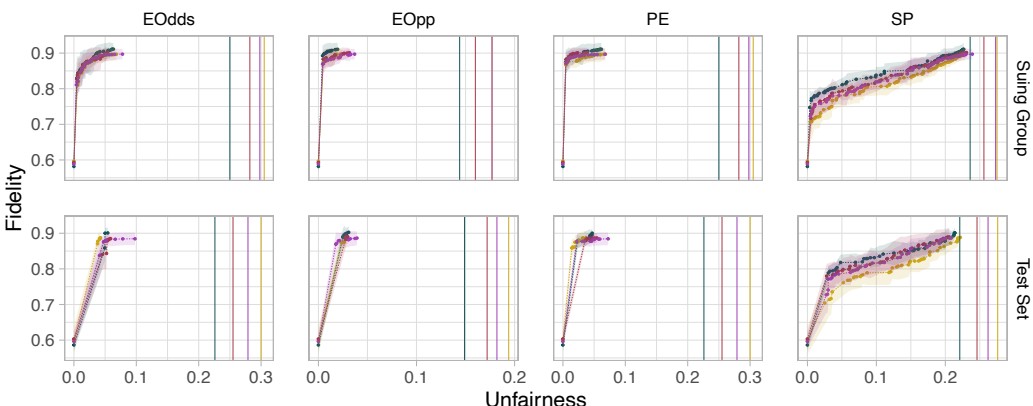

Figure 6: Fidelity-Unfairness trade-off of fairwashing attacks for equalized odds, equal opportunity, predictive equality and statistical parity metrics on COMPAS, using decision tree, logistic regression and rule list as explanation models. Vertical lines denote the unfairness of the black-box models. Results are averaged over 10 fairwashing attacks.

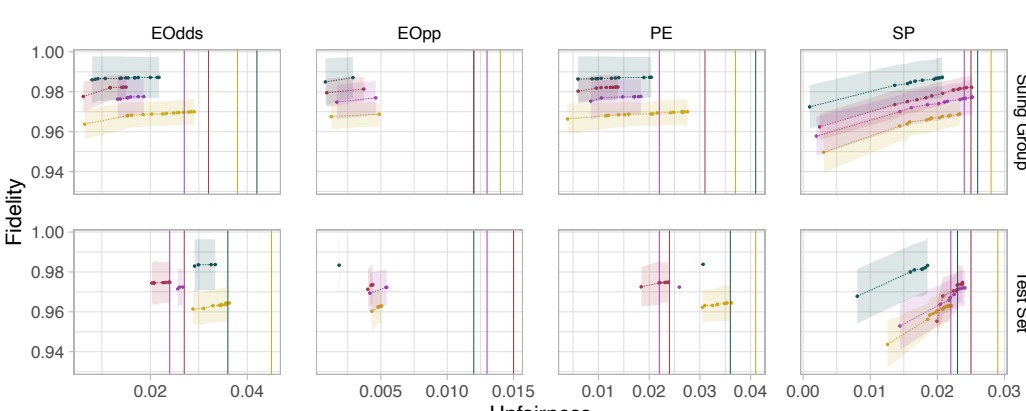

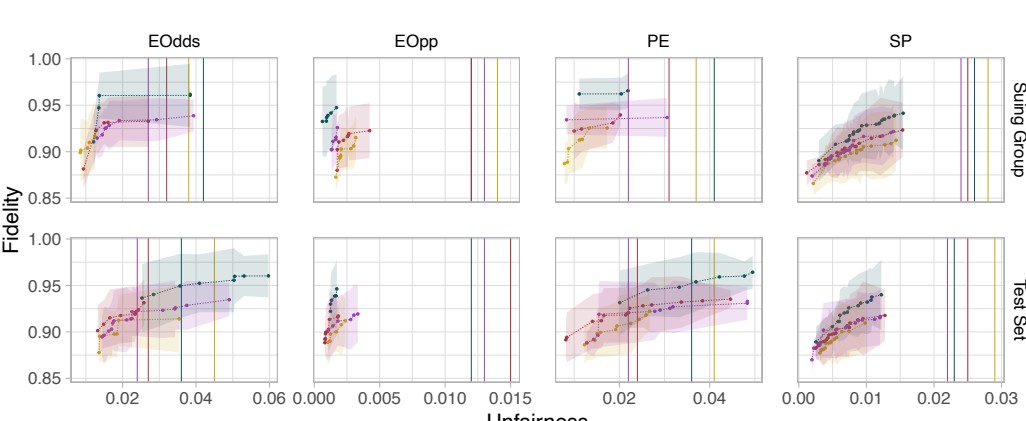

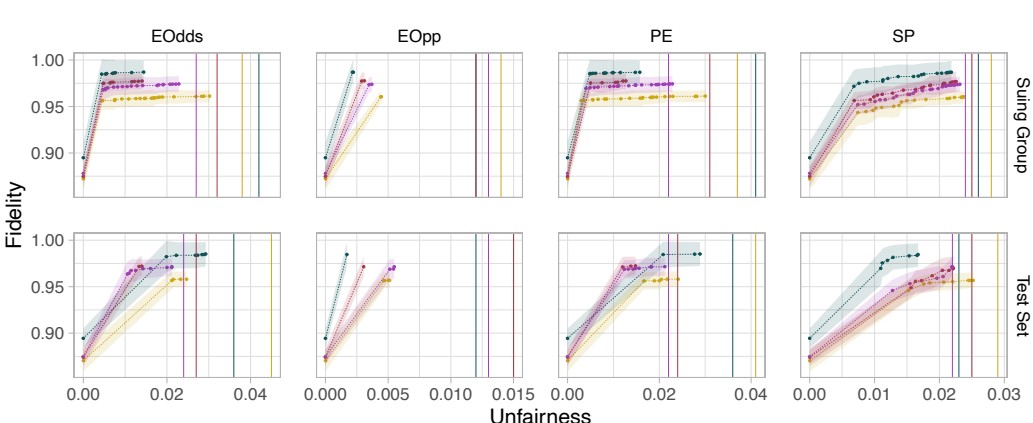

Figure 7: Fidelity-Unfairness trade-off of fairwashing attacks for equalized odds, equal opportunity, predictive equality and statistical parity metrics on Default Credit, using decision tree, logistic regression and rule list as explanation models. Vertical lines denote the unfairness of the black-box models. Results are averaged over 10 fairwashing attacks.

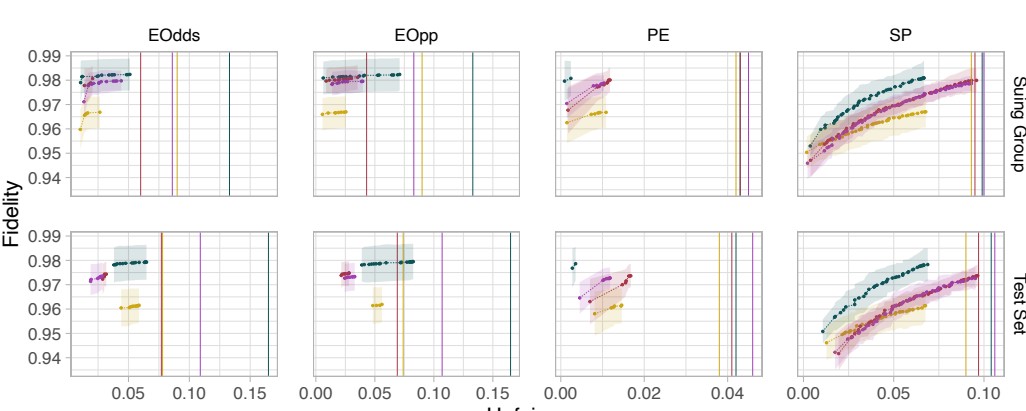

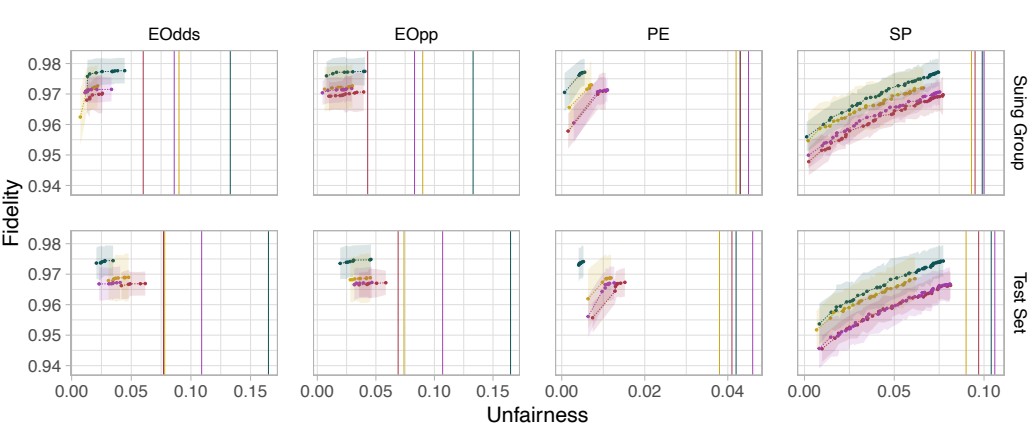

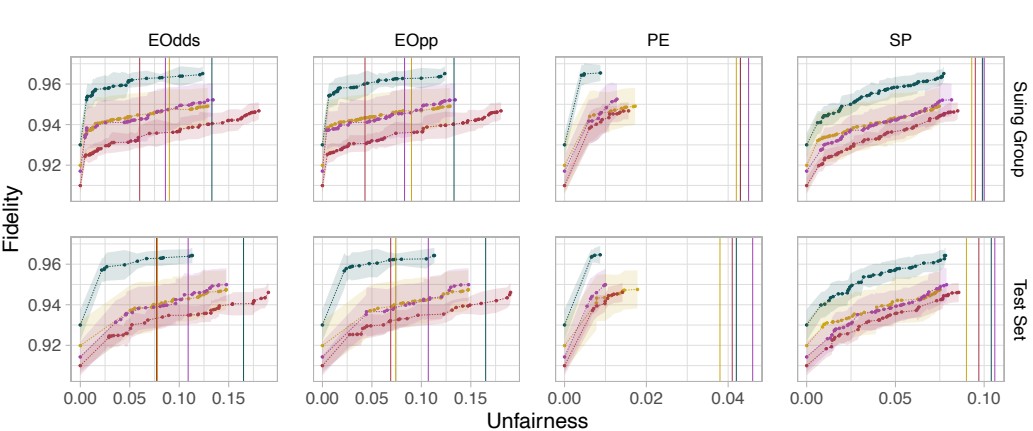

Figure 8: Fidelity-Unfairness trade-off of fairwashing attacks for equalized odds, equal opportunity, predictive equality and statistical parity metrics on Marketing, using decision tree, logistic regression and rule list as explanation models. Vertical lines denote the unfairness of the black-box models. Results are averaged over 10 fairwashing attacks.

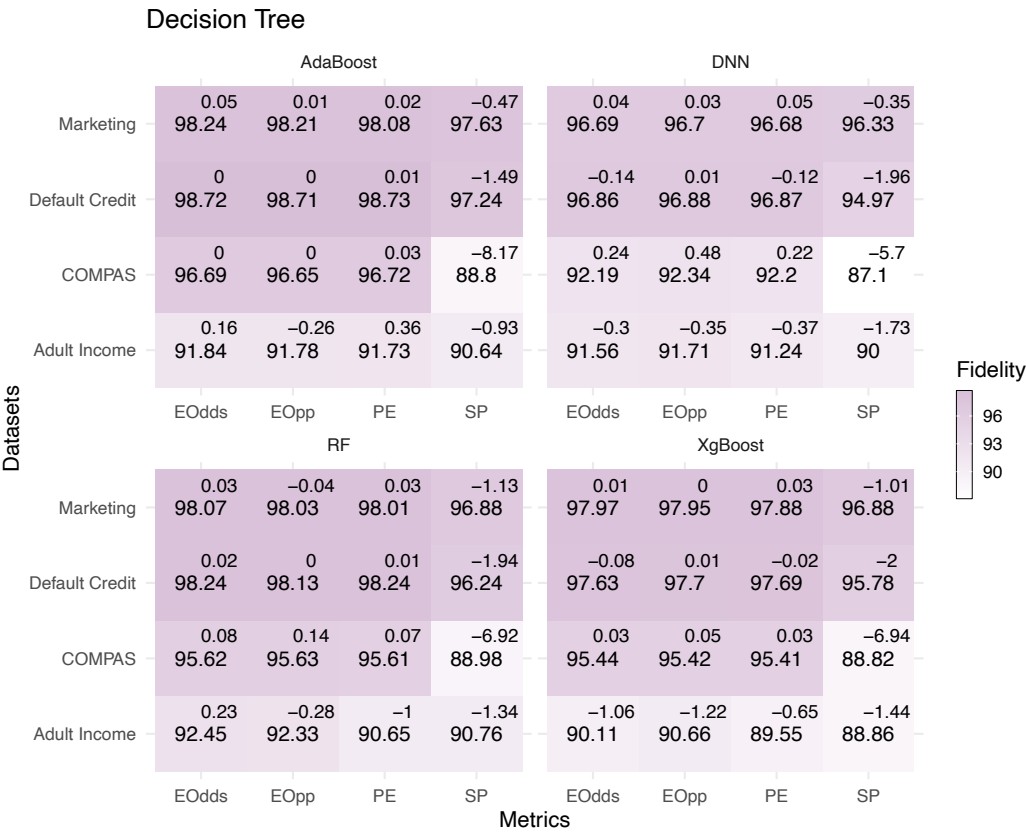

Figure 9: Fidelity of the fairwashed decision trees that are $50\%$ less unfair than the black-box models they are explaining. Results (averaged over 10 fairwashing attacks) are shown for all datasets, black-box models and fairness metrics. The content of each cell is in the form of $x^y$, in which $x$ represents the fidelity of the fairwashed explainer and $y$ its percentage change with respect to the fidelity of the unconstrained explanation model, used here as a baseline.

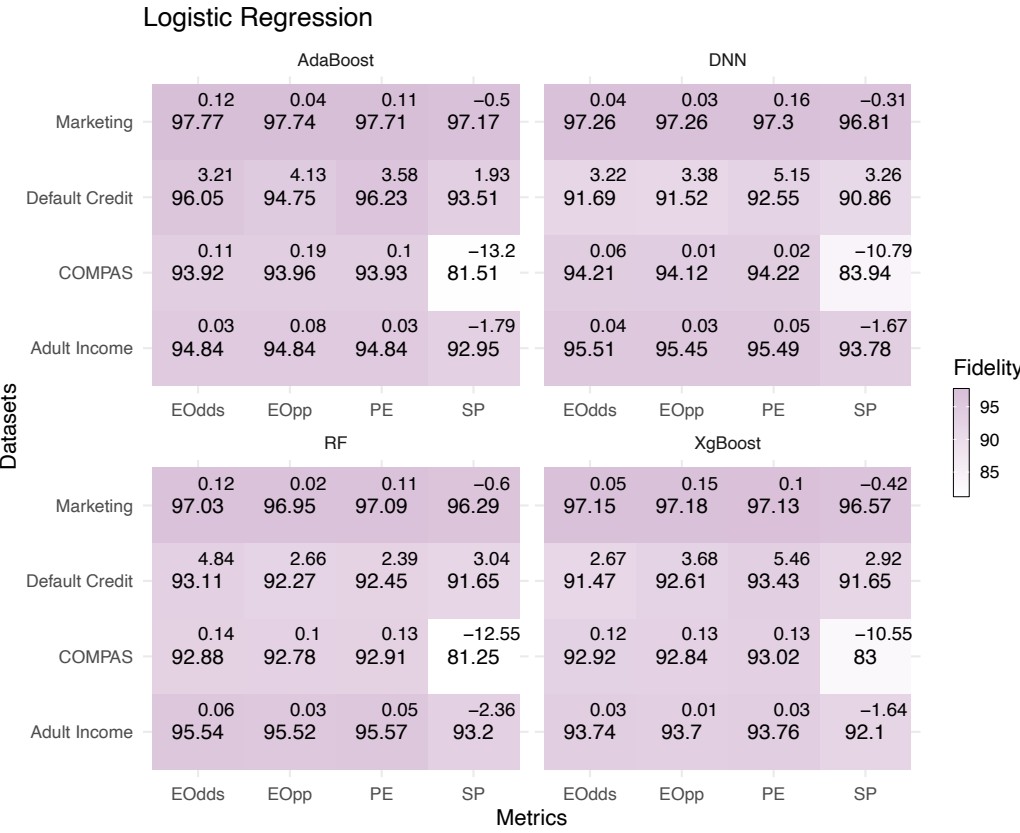

Figure 10: Fidelity of the fairwashed logistic regression models that are $50\%$ less unfair than the black-box models they are explaining. Results (averaged over 10 fairwashing attacks) are shown for all datasets, black-box models and fairness metrics. The content of each cell is in the form of $x^y$, in which $x$ represents the fidelity of the fairwashed explanation model, and $y$ its percentage change with respect to the fidelity of the unconstrained explainer, used here as a baseline.

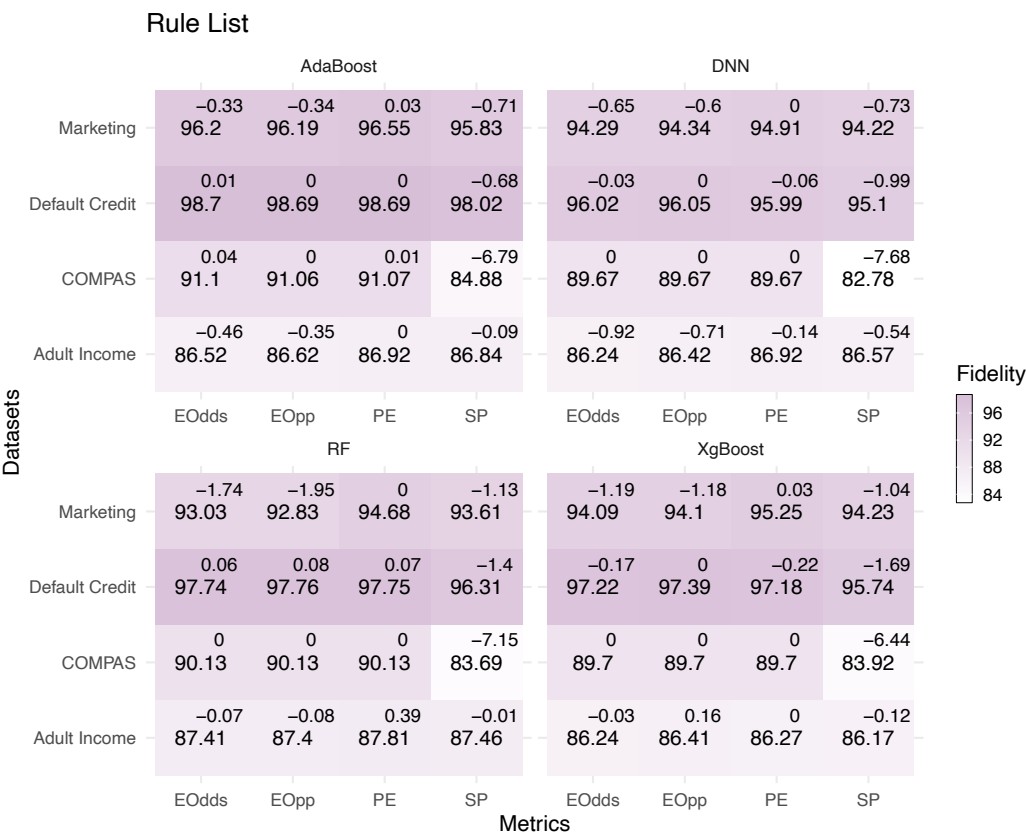

Figure 11: Fidelity of the fairwashed rule lists that are $50\%$ less unfair than the black-box models they are explaining. Results (averaged over 10 fairwashing attacks) are shown for all datasets, black-box models and fairness metrics. The content of each cell is in the form of $x^y$, in which $x$ represents the fidelity of the fairwashed explanation model, and $y$ its percentage change with respect to the fidelity of the unconstrained explanation model, used here as a baseline.

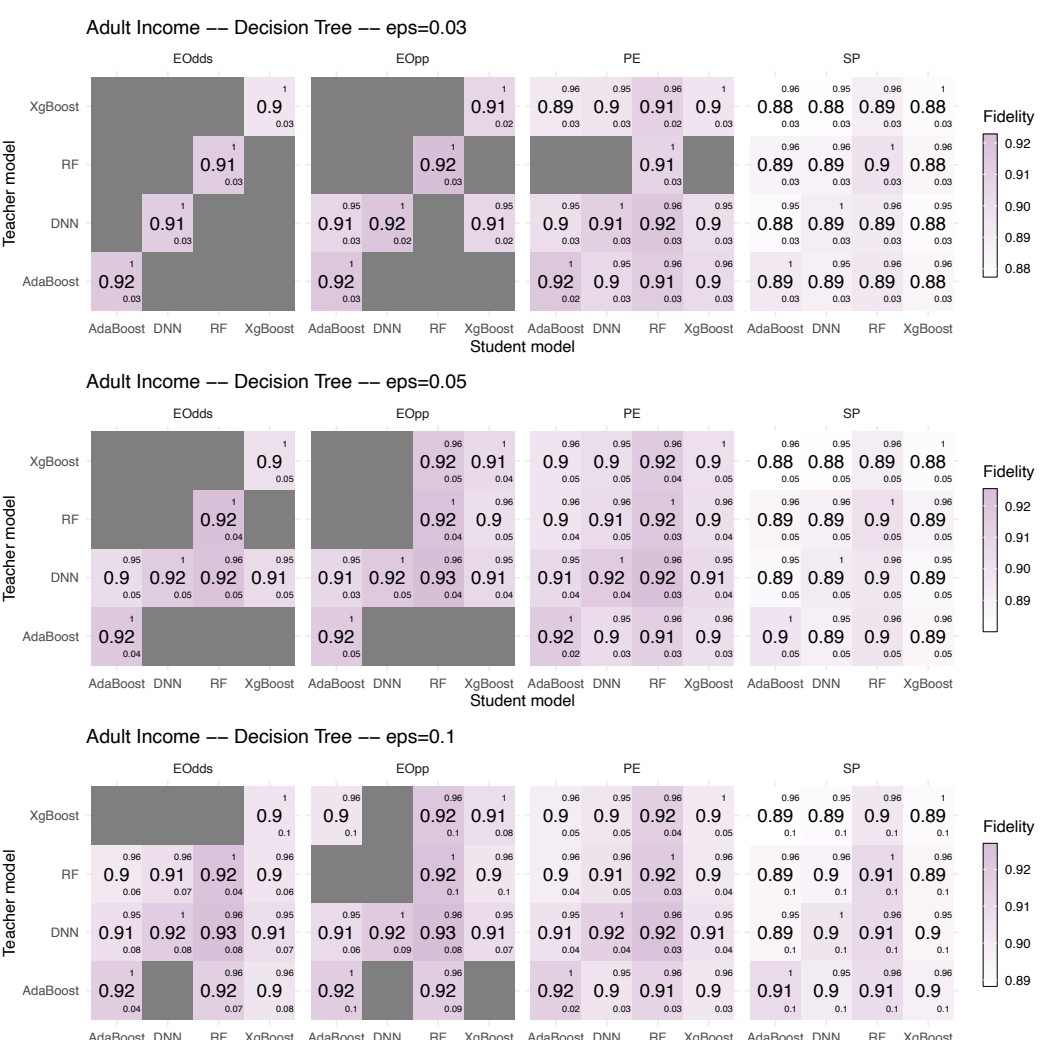

Figure 12: Analysis of the transferability of fairwashing attacks for equalized odds, equal opportunity, predictive parity and statistical parity on Adult Income, for different values of the unfairness constraint ($\epsilon \in \{0.03, 0.05, 0.1\}$), and for decision tree explanation models. The result in each cell is in the form of $x_z^y$, in which $y$ denotes the label agreement between the teacher black-box model and the student black-box model, $x$ is the fidelity of the fairwashed explanation model and $z$ is its unfairness. Blank cells denotes the absence of transferability for the unfairness constraint imposed. Results are averaged over 10 fairwashing attacks.

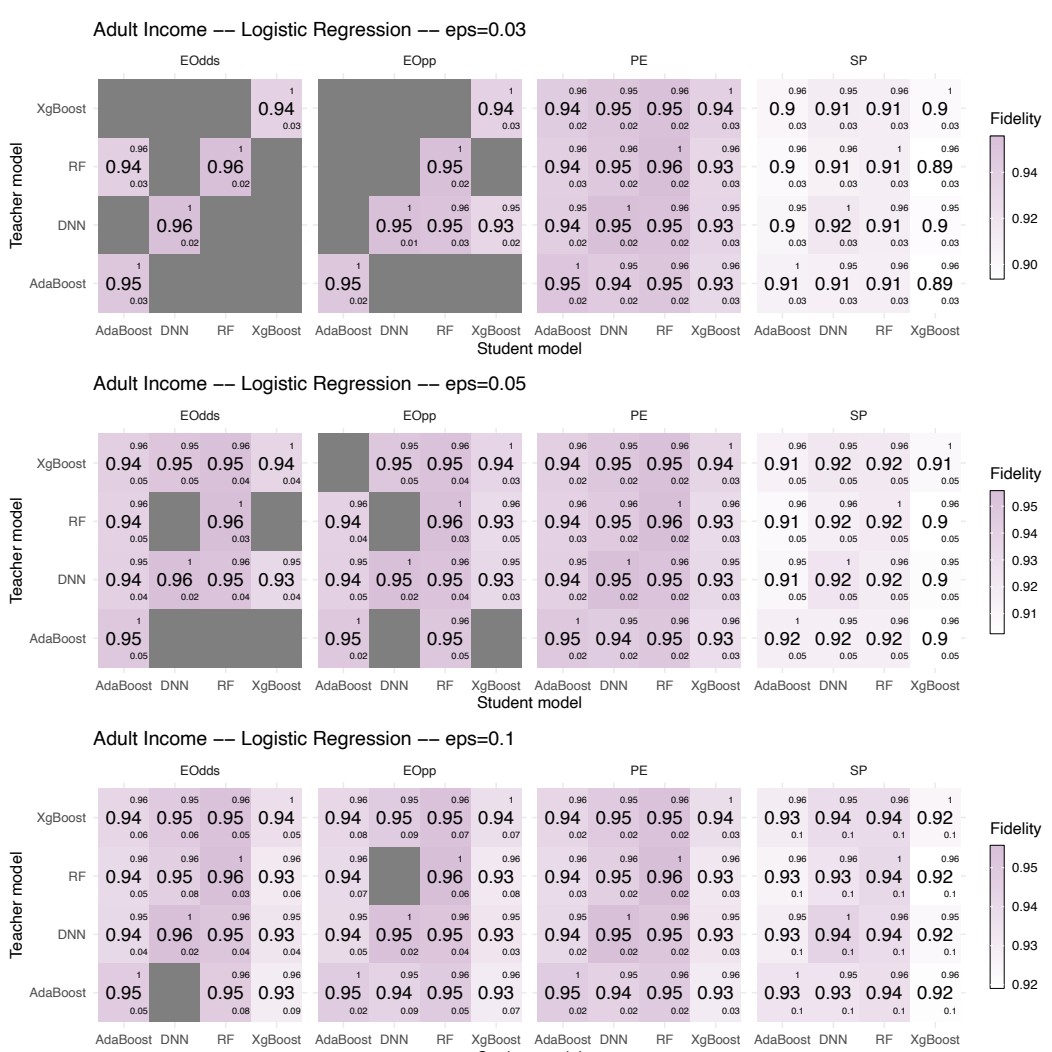

Figure 13: Analysis of the transferability of fairwashing attacks for equalized odds, equal opportunity, predictive parity and statistical parity on Adult Income, for different values of the unfairness constraint ($\epsilon \in \{0.03, 0.05, 0.1\}$), and for logistic regression explanation models. The result in each cell is in the form of $x_z^y$, in which $y$ denotes the label agreement between the teacher black-box model and the student black-box model, $x$ is the fidelity of the fairwashed explainer and $z$ is its unfairness. Blank cells denotes the absence of transferability for the unfairness constraint imposed. Results are averaged over 10 fairwashing attacks.

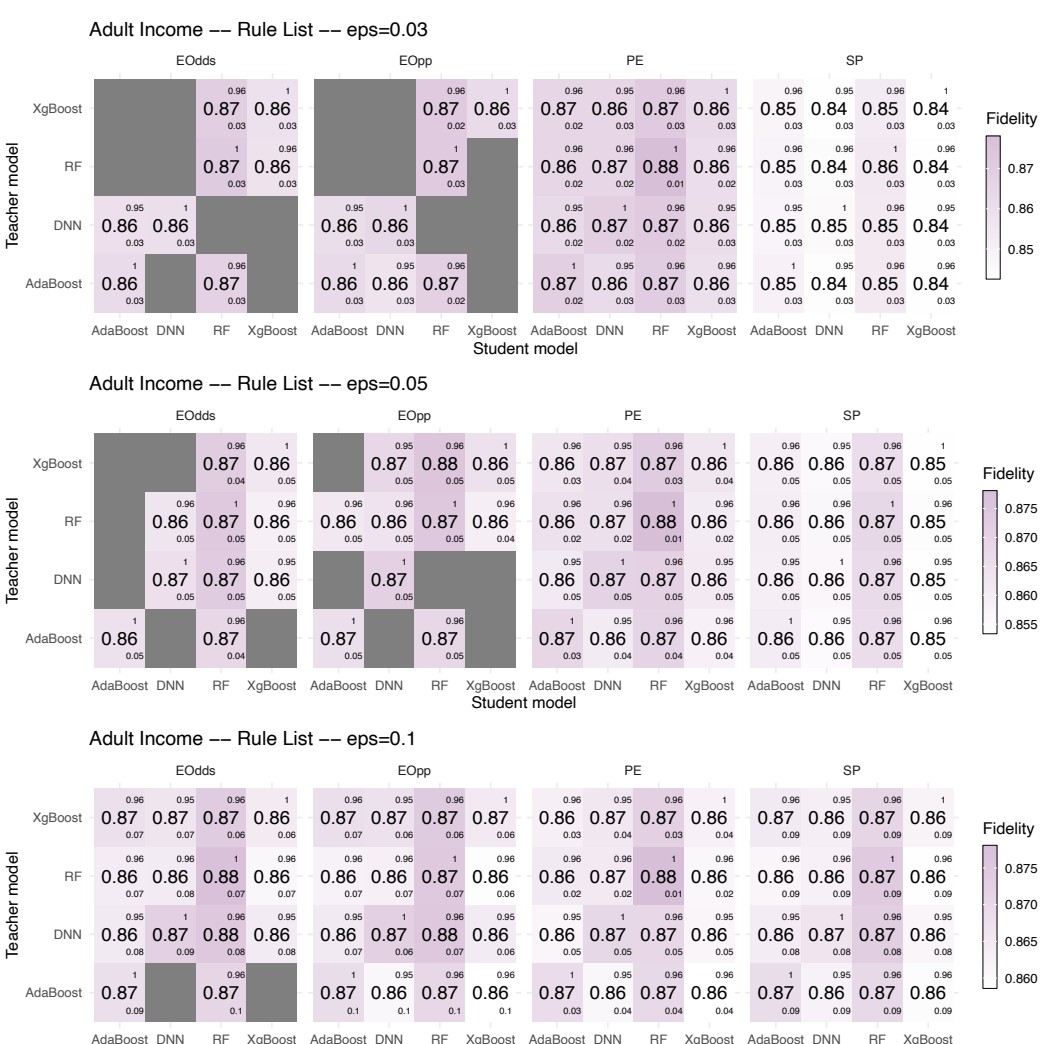

Figure 14: Analysis of the transferability of fairwashing attacks for equalized odds, equal opportunity, predictive parity and statistical parity on Adult Income, for different values of the unfairness constraint ($\epsilon \in \{0.03, 0.05, 0.1\}$), and for rule list explanation models. The result in each cell is in the form of $x_z^y$, in which $y$ denotes the label agreement between the teacher black-box model and the student black-box model, $x$ is the fidelity of the fairwashed explanation models and $z$ is its unfairness. Blank cells denotes the absence of transferability for the unfairness constraint imposed. Results are averaged over 10 fairwashing attacks.

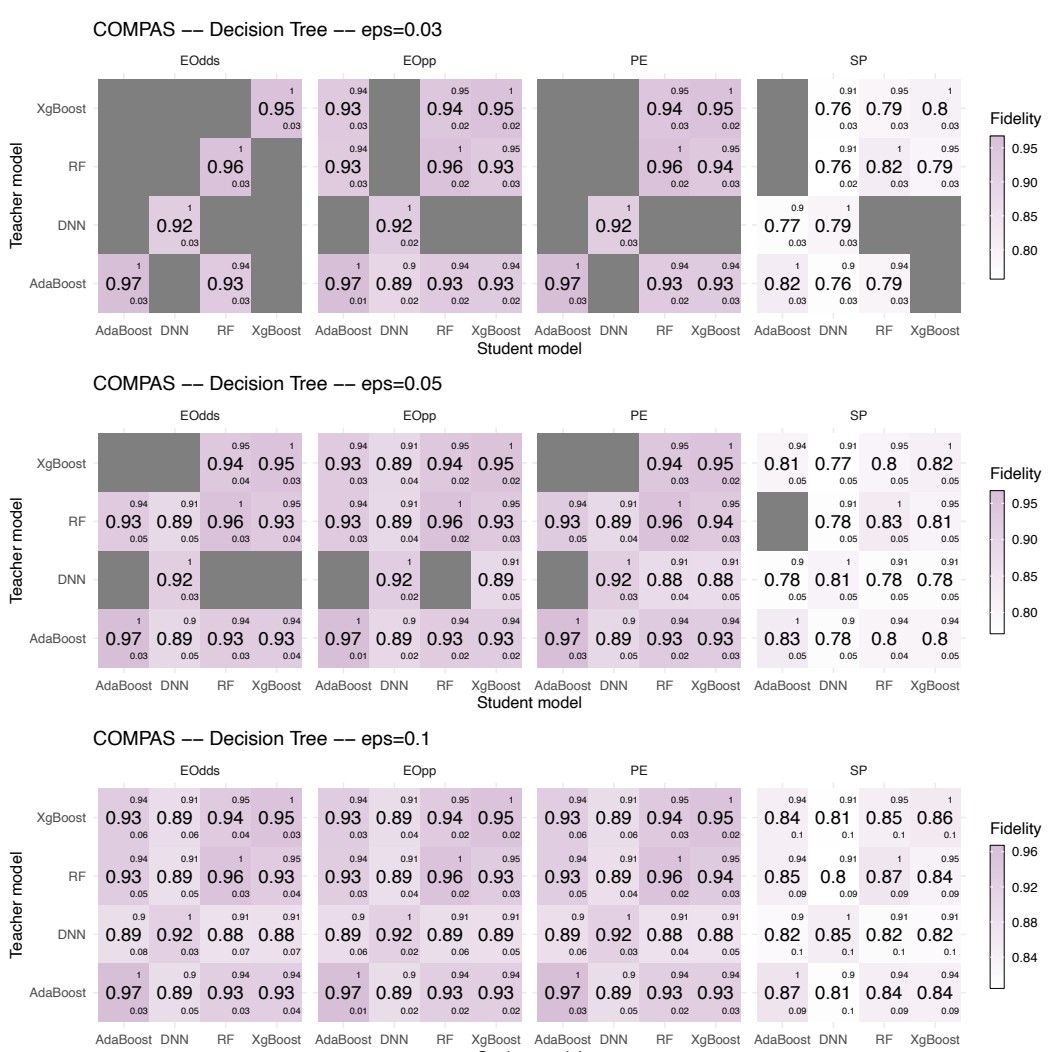

Figure 15: Analysis of the transferability of fairwashing attacks for equalized odds, equal opportunity, predictive parity and statistical parity on COMPAS, for different values of the unfairness constraint ($\epsilon \in \{0.03, 0.05, 0.1\}$), and for decision tree explanation models. The result in each cell is in the form of $x_z^y$, in which $y$ denotes the label agreement between the teacher black-box model and the student black-box model, $x$ is the fidelity of the fairwashed explanation model and $z$ is its unfairness. Blank cells denotes the absence of transferability for the unfairness constraint imposed. Results are averaged over 10 fairwashing attacks.

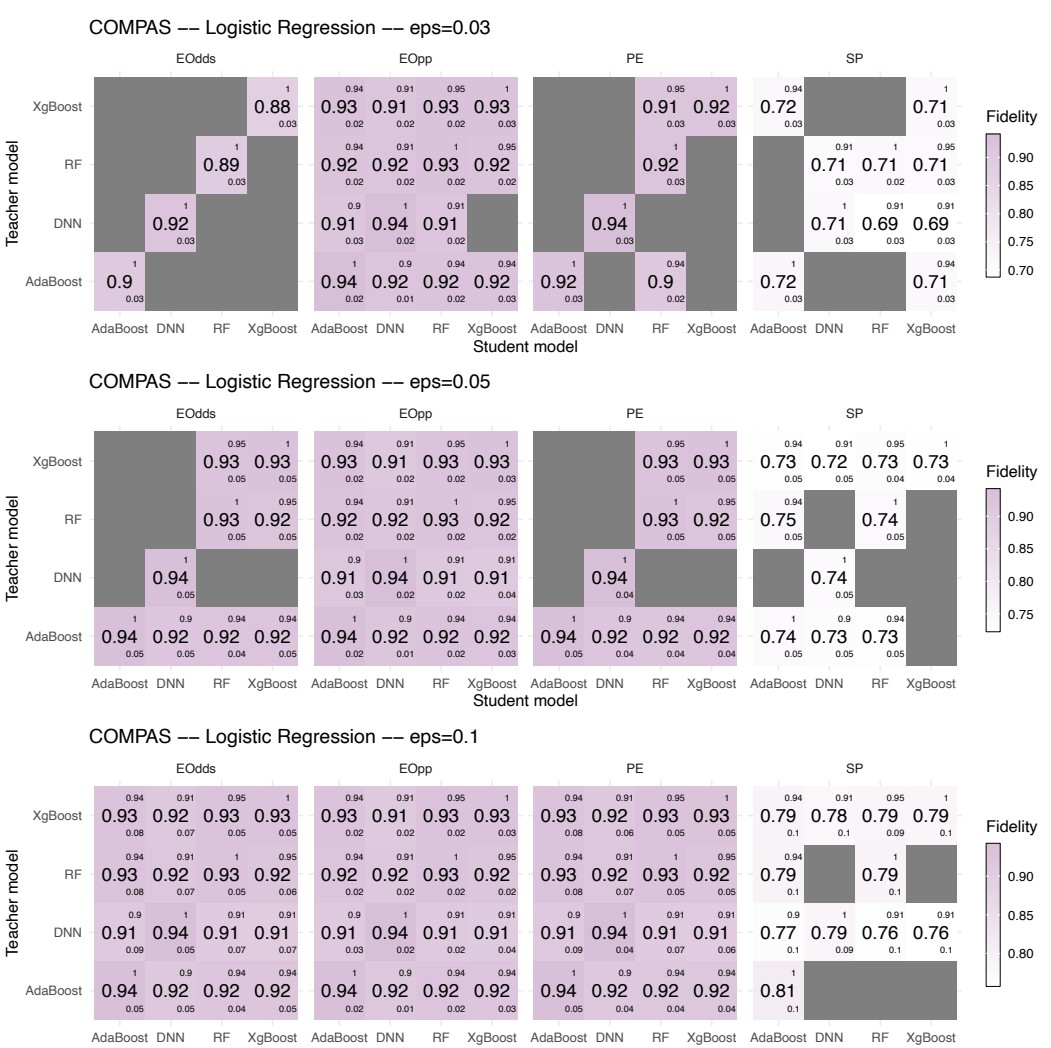

Figure 16: Analysis of the transferability of fairwashing attacks for equalized odds, equal opportunity, predictive parity and statistical parity on COMPAS, for different values of the unfairness constraint ($\epsilon \in \{0.03, 0.05, 0.1\}$), and for logistic regression explanation models. The result in each cell is in the form of $x_z^y$, in which $y$ denotes the label agreement between the teacher black-box model and the student black-box model, $x$ is the fidelity of the fairwashed explanation model and $z$ is its unfairness. Blank cells denotes absence of transferability for the unfairness constraint imposed. Results are averaged over 10 fairwashing attacks.

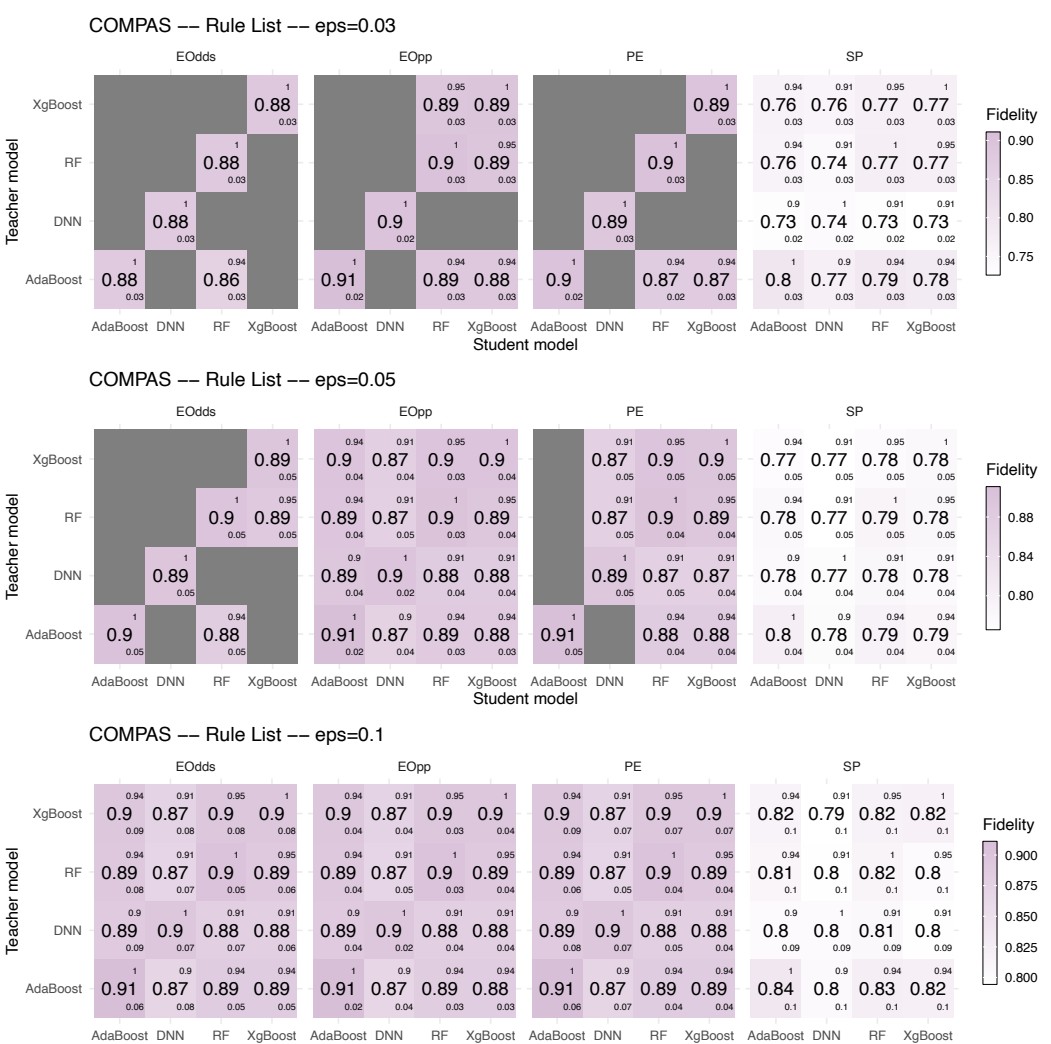

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

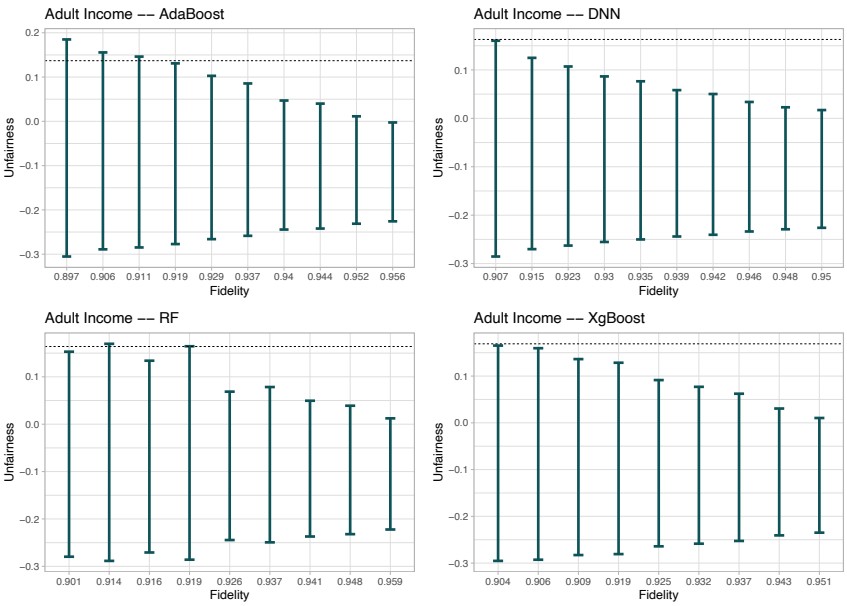

Figure 24: Range of the statistical parity of logistic regression explanation models for different values of the fidelity for AdaBoost, DNN, RF, and XgBoost black-box models trained on Adult Income. Horizontal lines denote the unfairness of the black-box models.

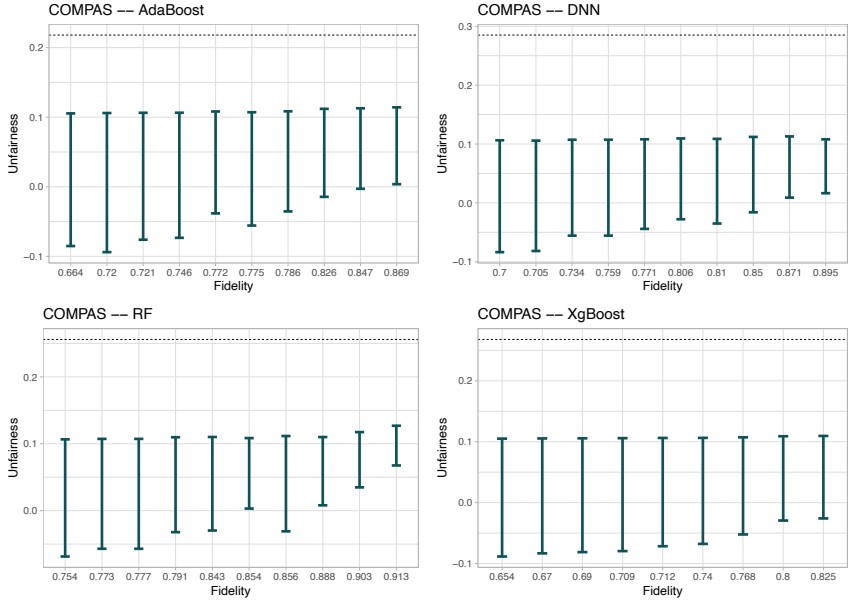

Figure 25: Range of the statistical parity of logistic regression explanation models for different values of the fidelity for AdaBoost, DNN, RF and XGBoost black-box models trained on COMPAS. Horizontal lines denote the unfairness of the black-box models.

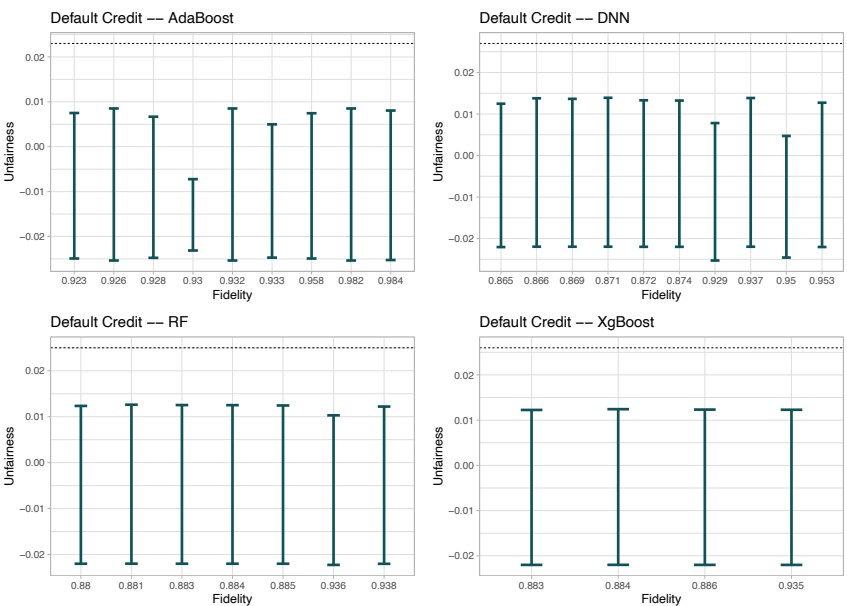

Figure 26: Range of the statistical parity of logistic regression explanation models for different values of the fidelity for AdaBoost, DNN, RF and XGBoost black-box models trained on Default Credit. Horizontal lines denote the unfairness of the black-box models.

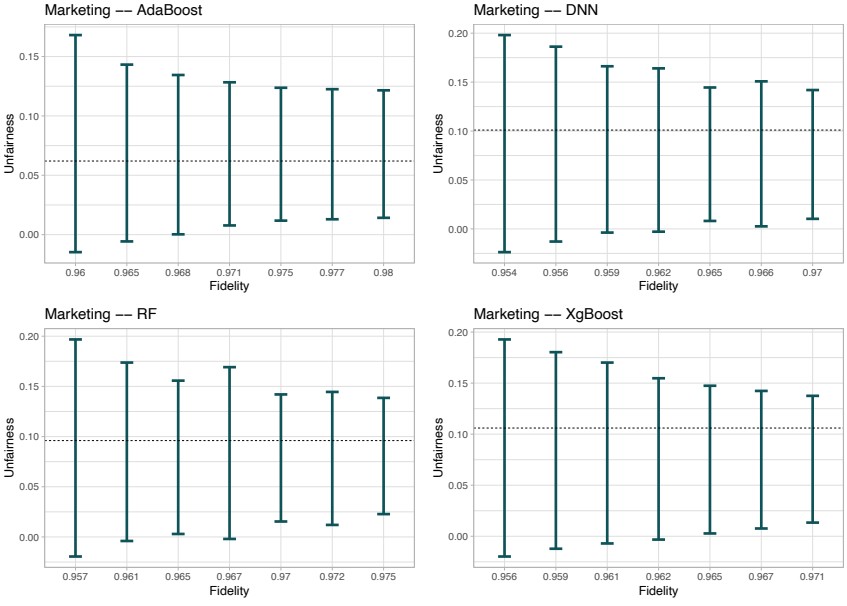

Figure 27: Range of the statistical parity of logistic regression explanation models for different values of the fidelity for AdaBoost, DNN, RF and XGBoost black-box models trained on Marketing. Horizontal lines denote the unfairness of the black-box models.