# OpenReview forum: "Characterizing the risk of fairwashing"
_NeurIPS.cc/2021/Conference — NeurIPS 2021 Poster_

### Official Review · Reviewer_Ga8q · 2021-07-01

**Rating:** 6
**Confidence:** 3

**Summary:**

This paper attempts to empirically characterize the risk of "fair washing" attacks, in which an unfair model can be described with high fidelity by a global explainer that exhibits substantially less unfairness. The authors determine the fidelity-unfairness Pareto curve for several black box models, explanation models, datasets, and fairness criteria and find that across all tested groups, it is generally possible to reduce apparent unfairness by 50% with a minimal reduction in fidelity to the black box model. The authors additionally show that fair washing attacks trained on a specific subgroup (the suing group) extrapolate well to an unseen test group and that fairwashed explanations trained on one black box model may still have high fidelity on other black box models. Finally, the authors determined the range of unfairness metric values that can be achieved at various levels of fidelity to the black box in order to quantify how susceptible different black box-dataset-explainer combinations may be to fair washing.

**Limitations And Societal Impact:**

I don't have any outstanding concerns about negative social impact after reading this paper.

**Main Review:**

The paper is well-written and begins to address the important issue of identifying and preventing fair washing attacks. My main concern is that the analysis involved mostly involves repeating tasks in the original fair washing paper [Aivodji 2019] on a wider range of black box/explainer/dataset/metric combinations, without attempting to rigorously quantify the results and how they may relate to the characteristics of the black box/explainer/dataset/metric combination.

In particular, experiment 1 generally seems to repeat the primary result of [Aivodji 2019], but on a larger combination of black box/explainer/dataset/metrics. I would find this more interesting (and more reflective of the title "characterizing" the risk of fair washing) if it were possible to determine some properties or trends across these results: e.g. how and why are the results for COMPAS different than those for default credit? How and why are the results for LogReg different than those for rule lists?

Again, experiment 2 repeats a result from [Aivodji 2019], but on a larger combination of black box/explainer/dataset/metrics.

Experiment 3 is new relative to [Aivodji 2019], but I find the motivation rather unconvincing, and perhaps the result is unsurprising - if the signal in the dataset is simple, or if black box accuracy is high across several black boxes, it mostly seems natural that the explanation for one will be a reasonably good match for another, with some variation around decision boundaries.

Experiment 4 again uses a method from another work [Coston 2021] in a very straightforward way, without attempting to characterize the variation in results.

Overall, I think the work is well-presented, but I find that it is lacking with respect to some significant contribution toward the understanding of when and why fair washing attacks are possible.

Clarity-wise, my only complaint is that the variation in black box/explainer/dataset/metric combinations presented in the main paper images is somewhat hard to follow, e.g. Figures 1, 2, and 3 concern Decision Tree, Figure 4 concerns Logistic Regression. Perhaps either remain consistent or provide some rationale for the switch.

**Time Spent Reviewing:**

4

---

> ### Author Response · Authors · 2021-08-10
> **Author response**
>
> We thank the reviewer for the detailed review. Our responses to the questions are below. We would be grateful if the reviewer would consider raising the score in light of our responses.
>
> ---
> “Overall, I think the work is well-presented, but I find that it is lacking with respect to some significant contribution toward the understanding of when and why fair washing attacks are possible.”
>
> The main contributions of our paper are the study of the (un)detectability of fairwashing and the quantification of the risk of fairwashing. The fundamental question regarding fairwashing attacks is whether we can detect them. In this work, we provide the first results demonstrating clearly that detecting manipulated explanations will not be possible (or at least very difficult). Specifically, our results show that the two types of inconsistency of explanations below are not helpful for detecting manipulated explanations.
>
> * Inconsistency between different suing groups.
>
> In a fairwashing attack, a manipulated explanation is tailored specifically for a suing group of interest so that the explanation is fair when evaluated on the suing group. Because the explanation is specific to that group, there is a possibility that the same explanation can fail on another suing group. One can use this fact to detect the explanation’s manipulation. That is, if there is a second suing group that is not disclosed to the model providers, one can apply the provided explanation to that second group and observe whether the explanation is fair or not. Assume now that the provided explanation is unfair on the second group. In that case, the explanation is likely to be a manipulated one, tailored specifically to the first suing group disclosed to the model producers.
>
> * Inconsistency between different models.
>
> In a fairwashing attack, a manipulated explanation is designed specifically for the deployed black-box model. However, in practical machine learning, it is usually the case that the deployed model is updated frequently. This will lead to an inconsistency between the manipulated explanations provided to the suing group in the past and the currently deployed model. If the decisions of the currently deployed model largely conflict with the past explanations, there is a possibility that the current model is unfair or that the past explanation was manipulated.
>
> Experiments 2 and 3 revealed that the two inconsistencies discussed above are small on several datasets, black-box models, and fairness criteria. These results indicated that these inconsistencies are not helpful for detecting fairwashing attacks in practice. Thus it will not be possible to determine if the provided explanation has been manipulated with high confidence.
>
> ---
>
>  “In particular, experiment 1 generally seems to repeat the primary result of [Aivodji 2019], but on a larger combination of black box/explainer/dataset/metrics.”
>
> We acknowledge that Experiment 1 is an in-depth analysis of the main result of [Aivodji 2019] on a larger combination of black box/explainer/dataset/metric. The main objective of this experiment is to provide a full characterization of the fidelity-unfairness Pareto frontier of a fairwashing attack. On a technical level, this experiment uses an epsilon-constraint approach to produce a fine-grained fidelity-unfairness Pareto frontier in comparison to the work of [Aivodji et al. 2019], in which a regularized rule list enumeration algorithm was used. To avoid confusion on this work's main contributions, we proposed removing Experiment 1 from the list of contributions and use it only as a warm-up and motivation case. The main contributions of the paper are the study of the (un)detectability of fairwashing attacks and the quantification of the risk of fairwashing.
>
> ---
> “Again, experiment 2 repeats a result from [Aivodji 2019], but on a larger combination of black box/explainer/dataset/metrics.”
>
> In [Aivodji et al. 2019], concerning the generalization of fairwashing beyond the suing group, the authors said: “Our preliminary experiments in (S1) show that the fidelity of the model rationalization on the test set tends to be slightly lower than the one on the suing group, which means that the explanation is customized specifically for the suing group.” This remark suggested that analyzing the performances of an explainer on unseen data can help in detecting fairwashing. In this paper, we conduct an extensive empirical study to show that this is not quite the case as we demonstrated that fairwashed explainers generalize well beyond suing groups. Our new findings suggest that using unseen data to evaluate the performances of explainers is not a viable strategy to detect fairwashing. This experiment and its findings also aim to raise the awareness that fairwashed explainers can be produced quite easily.
>
> ---
>
> “Experiment 3 is new relative to [Aivodji 2019], but I find the motivation rather unconvincing, and perhaps the result is unsurprising...”
>
> We agree that it is not surprising to have transferability because accuracy is high across several black-boxes. The point of this experiment, in addition to showing the difficulty of detecting fairwashing, was to emphasize on how the multiplicity of good models (high-accuracy black-box models), which is inevitable in most real-world datasets as demonstrated by several recent studies (e.g., [D’Amour et al. 2020]), turns out to be a great enabler for the transferability of fairwashed explainers, making them even cheaper to produce.
>
> ---
>
> “Experiment 4 again uses a method from another work [Coston et al 2021] in a very straightforward way, without attempting to characterize the variation in results.”
>
> The results of Experiments 2 and 3 reinforce the intuition that the evaluation of the risk of fairwashing should go beyond specific data instances and model class, by also taking into account the prediction task considered. To this end, the main rationale of Experiment 4 is precisely to view the risk of fairwashing as something that also depends on the prediction task. As such, practitioners can perform it to assess, given the dataset at hand, the variability of the unfairness of high-fidelity explainers. This variability will, in return, inform on the risk of fairwashing.
>
> ---
> “Clarity-wise, my only complaint is that the variation in black box/explainer/dataset/metric combinations presented in the main paper images is somewhat hard to follow, e.g. Figures 1, 2, and 3 concern Decision Tree, Figure 4 concerns Logistic Regression. Perhaps either remain consistent or provide some rationale for the switch.”
>
> We thank the reviewer for the remark. We will add and present the results with logistic regression across all four experiments to remain consistent.

---

> > ### Comment · Reviewer_Ga8q · 2021-08-11
> > **Thanks for response, consider updating your narrative**
> >
> > My understanding (generally, and from the framing of the paper) is that the "risk of fairwashing" suggests quantifying the likelihood that a given problem setting (black box/explainer/dataset/metric) will admit a fairwashing attack, which I maintain isn't exactly addressed here. While there are some nice empirical results, there's not much information on how to generalize these results to an unseen setting.
> >
> > That said, if the focus of your paper is indeed intended to be "characterizing the undetectability and risk of fairwashing", then experiments 2 and 3 seem more applicable. I don't think the current framing of the paper reflects this clearly. If you intend to reframe the paper in this manner, I can increase my score.
> >
> > In Experiment 2, it would be useful to point out and contextualize the difference between your results and that which was hypothesized in Aivodji et al 2019.

---

> > > ### Author Response · Authors · 2021-08-11
> > > **Thank you for your understanding**
> > >
> > > We are very grateful to the reviewer for increasing the score in light of our clarifications and the reframing we suggest. In the revised version of the paper, we will indeed point out and contextualize the difference between the result of Experiment 2 and that which was hypothesized in Aivodji et al 2019.

---

### Official Review · Reviewer_KFAW · 2021-07-04

**Rating:** 7
**Confidence:** 4

**Summary:**

This paper studies "fairwashed" explanation of black-box ML models -- interpretable models that achieve high fidelity with respect to the black-box predictions but substantially better scores on various fairness metrics. The authors empirically demonstrate how fairwashing can be achieved with very small decreases in fidelity, and how a fairwashed explanation of one type of model (say, random forests) can be successfully repurposed to explain a different model (eg AdaBoost). Because of this, they argue, auditors need to be very careful about how they use interpretable models in fairness assessments.

**Limitations And Societal Impact:**

This paper is focused on the bad outcomes that could occur when interpretable models are used to assess fairness claims. It is clearly concerned with averting the potential negative societal impact of such assessments.

**Main Review:**

This is a well-written, interesting paper about an area of growing concern for the field. Its goals -- to characterize what is possible with fairwashing attacks -- are reasonably modest, but it lands its point clearly. There's room for expanding on this work, but I think it would be a worthy addition to NeurIPS.

### Questions

Why not also test calibration on the continuous predictions, since this is very often what is used to measure fairness in real-world settings?

Intuitively, it seems like these attacks would be easy to identify because to decrease some unfairness measure by 50% while only changing <1% of predictions would require obvious structure in the changed predictions (eg all the changes affect the same group). Is this the case?

### Notes

The x axes in Fig 1 would be more readable if they didn't have the same scale.

**Time Spent Reviewing:**

1

---

> ### Author Response · Authors · 2021-08-10
> **Author response**
>
> We thank the reviewer for the detailed review and for recommending the acceptance of our paper. Our responses to the questions are below.
>
> ---
>
> “Why not also test calibration on the continuous predictions, since this is very often what is used to measure fairness in real-world settings?”
>
> All the explainers that we used are probabilistic classifiers and as such, they can all be calibrated. In this paper, we used an in-processing approach to design the constrained explainers. However, both pre-processing and post-processing methods can be used as well. Additionally, we would like to point out that the only practical implementation of fairness calibration in the literature is calibrated equalized odds post-processing Pleiss et al. (NeurIPS 2017), which uses a generalized definition of equalized odds. We have also considered equalized odds as a fairness metric in this work and show that fairwashing can apply to this family of metrics.
>
> ---
>
> “Intuitively, it seems like these attacks would be easy to identify because to decrease some unfairness measure by 50% while only changing <1% of predictions would require obvious structure in the changed predictions (e.g., all the changes affect the same group). Is this the case?”
>
> Actually, this is not quite the case. We compute the gap between group-level fidelity values for each fairwashed explainer. Overall, one can see that the gap is very small (less than 2% in most cases). We will provide a graph that shows the gap as a function of the explainer’s unfairness in the appendix of the revised version of the paper.
>
> ---
>
> “The x axes in Fig 1 would be more readable if they didn't have the same scale.”
>
> Thank you very much for the suggestion. We will improve the clarity of the graphs by freeing the x-axes in Figure 1.
>
> ---
> Limitations And Societal Impact: “This paper is focused on the bad outcomes that could occur when interpretable models are used to assess fairness claims. It is clearly concerned with averting the potential negative societal impact of such assessments.”
>
> Thank you for the positive assessment of the societal impact of our work.

---

> > ### Comment · Reviewer_KFAW · 2021-08-23
> > **Response to authors**
> >
> > Thank you for your response.
> >
> > >"the only practical implementation of fairness calibration in the literature is calibrated equalized odds post-processing Pleiss et al."
> >
> > There are plenty of calibration approaches in the ML literature (eg Platt scaling, isotonic regression). These aren't specifically confined to the fairness literature because they have other uses and pre-date ML fairness efforts.
> >
> >
> > >"We compute the gap between group-level fidelity values for each fairwashed explainer."
> >
> > Even if group level fidelities are equal, there could still be obvious structure in the changed predictions, right? If <1% of predictions are changed then group level fidelities must differ by <2% (assuming equal sized groups), but it could still be the case that _the changed predictions_ exhibit obvious structure (eg all flipped predictions affect the same group, or all positively-flipped predictions affect one group and all negatively-flipped predictions affect another).

---

> > > ### Author Response · Authors · 2021-09-02
> > > **Thank you for your comment**
> > >
> > > We thank the reviewer for the response. Our responses to the questions are below.
> > >
> > > **Comment on calibration**
> > >
> > > We thank the reviewer for the clarification. We believe that any measurable property can be “washed”. That is, having an explainer that exhibits positive values for the property of interest while it is not the case for the original black-box model. Adding calibration is doable. However, we leave this task to future work as it is not applicable straightforwardly.
> > >
> > > **Comment on fidelity**
> > >
> > > We thank the reviewer for the clarification and this very interesting question. We performed a preliminary experiment on the four datasets to evaluate the behaviour of the fidelity at a group level for each label. Overall, we noticed some disparities across labels and groups that hold for both fairwashed and non-fairwashed explainers. However, the results are not exactly the same. So, one can train a meta-classifier to distinguish between fairwashed and non-fairwashed explainers using those features. However, such a task would require several pairs of fairwashed explanations and honest explanations and is beyond the scope of the current paper. We will mention this in the Future Work section of the paper.

---

### Official Review · Reviewer_qEv3 · 2021-07-16

**Rating:** 6
**Confidence:** 4

**Summary:**

This paper addresses the problem of fairwashing, i.e., the process in which an adversary builds a manipulated explanation of the unfair black-box model. The manipulated explanations should ideally hide the fact that the block-box model is unfair. The authors conduct a series of experimental evaluations and conclude:
-	Manipulated explanations can be up to 99.2 % as accurate as the original model, while being much less (up to 50% less) unfair
-	Fairwashed explanations can generalize beyond the data points they were trained on.


**Limitations And Societal Impact:**

The limitations of the proposed fidelity measure could be better discussed and evaluated (see comments above).

**Main Review:**

The paper addresses a timely and relevant topic in the area of model accountability. However, there are several concerns regarding this work:
1. The authors conduct their experimental evaluation for one fair-washing attack; yet the title suggests that a general approach has been identified.
2. The experimental results in the main paper are shown for one data set only. Yet, there is plenty of space dedicated to a table summarizing related work. I suggest presenting some of the experimental results from the appendix in the main paper.
3. I do not believe that the way fidelity is described in the paper provides a useful measure to understand how faithful the explanation model is to the black-box model. As an example, an explanation model with a probability of 0.51 would receive the same fidelity score as a model with a probability of 1. Moreover, even when the probability is 1 (or 0), the pre-sigmoid scores (i.e., logits) can still be substantially different, essentially indicating that the explanation model reasons differently. In principle one could imagine that the former explanation model, when outputting a constant class, would be much fairer than a more discriminative model.
3. At times, the paper is not clear! I will give examples below.

Clarifications & Questions:
-	The authors have not really made clear what their paper is trying to achieve: sometimes it sounds as if developing more stable methods is a concern?
-	Question: Figure 1 - EOpp is essentially constant across different unfairness thresholds. In contrast to that, the fidelity under PE and SP constraints are more responsive to the fairness thresholds. Why is that? Has it to do with EOpp not considering individuals with true label y=0?
-	How does the proposed method generalize for the attacks by Dombrowski et al (2020, ICML) and Slack et al (2020, AIES)? Can the authors comment on this?
-	At no point, the suing group X_sg was formally defined. What kind of data points are in there? Are those random samples from the data set?
-	Equation 1 contains very little to no intuition. It would have been nice to include some sentences describing the objective and why we should care about it.
-	There are many claims being made which I feel are unsubstantiated. At least, there is no theory supporting such strong claims, e.g.,
 in the Abstract: ‘[…] show that fairwashed explanation models can generalize beyond the suing group, which will only worsen as more stable fairness methods get developed.’ To me this seems like speculation.
-	In the figures, the vertical lines are thin and hard to distinguish.
-	In Section 3.1, you first describe your experimental results in Figures 1 and 2. Then, in a separate paragraph, you describe the conclusions without referring to the figures these conclusions could be based on.


**Time Spent Reviewing:**

4

---

> ### Author Response · Authors · 2021-08-10
> **Author response**
>
> We thank the reviewer for the detailed review. Our responses to the questions are below. We would be grateful if the reviewer would consider raising the score in light of our responses.
>
> ---
>
> “The authors conduct their experimental evaluation for one fair-washing attack; yet the title suggests that a general approach has been identified.”
>
> With respect to the adequation between the title and the experimentation conducted, as we indicated in our paper (Line 50), we focused on the class of global explanation techniques. For this particular class, only two works in the literature have studied explanation manipulation: Aïvodji et al. (ICML 2019) and Lakkaraju & Bastani (AIES 2020). The former relies on quantifiable fairness metrics to describe the “bad behaviour” that one seeks to hide, while the latter relies on legitimate features. We leverage the work of Aïvodji et al. (ICML 2019) because it is more generic than Lakkaraju & Bastani (AIES 2020) as it can handle both direct and indirect bias. Overall, the approach we used is general enough for the whole class of global explanation techniques, in which one seeks to explain the behaviour of a black-box model with a single interpretable model trained on a set of data instances. To be more specific, if possible, we proposed changing the title of our paper to “Characterizing the risk of fairwashing in global explanation techniques''.
>
> ---
> “The experimental results in the main paper are shown for one data set only. Yet, there is plenty of space dedicated to a table summarizing related work. I suggest presenting some of the experimental results from the appendix in the main paper.”
>
> Following the reviewer's suggestion, we will move more experimental results from the appendix to the main paper by using the space of the related work summary table.
>
> ---
>
> “I do not believe that the way fidelity is described in the paper provides a useful measure to understand how faithful the explanation model is to the black-box model….”
>
> We totally agree with the reviewer that fidelity is not a good proxy for the quality of an explanation. In fact, this is clearly one of the main messages of our paper. However, to raise awareness on the risk of manipulation, we have to consider metrics that are well established in the community and demonstrate in a quantifiable way how they can be unreliable to foster the design of better metrics. The definition of fidelity that we considered in the paper was first introduced in Craven and Shavlik (NeurIPS 1996) and is still the main quality criterion used in global explanation techniques.
>
> ---
>
> “There are many claims being made which I feel are unsubstantiated. At least, there is no theory supporting such strong claims, e.g., in the Abstract: ‘[…] show that fairwashed explanation models can generalize beyond the suing group, which will only worsen as more stable fairness methods get developed.’ To me this seems like speculation.”
>
> We acknowledge that the expression “... which will only worsen as more stable fairness methods get developed” may give the false impression that we are saying that developing stable methods is a concern in general. We want to clarify that this is absolutely not what we meant. Instead, we wanted to point out that an adversary can leverage to his benefit stable learning methods to devise more powerful fairwashing attacks that generalize beyond the suing group. Fundamentally, the fairwashing attack that we considered can be recast as the problem of training an interpretable model (i.e., the explainer) under unfairness constraint, using a suing group as the training set (more precisely, the reduction of a fairwashing attack to a fair learning task is explained in Line 241 of the paper). Thus, generalizing beyond this particular suing group is the equivalent of generalizing beyond the training set in a fair learning problem.
>
> ---
>
> “Figure 1 - EOpp is essentially constant across different unfairness thresholds. In contrast to that, the fidelity under PE and SP constraints are more responsive to the fairness thresholds. Why is that? Has it to do with EOpp not considering individuals with true label y=0?”
>
> Following the suggestion of Reviewer KFAW, we have created a more readable version of Figure 1 by freeing the x-axes. The trade-offs for PE and EOpp now look very similar. Additionally, it is important to note that sometimes, even an unconstrained explainer is significantly fairer than the black-box model for some of the fairness metrics. As a result, the slope of the trade-off curve is small.
>
> ---
>
> “How does the proposed method generalize for the attacks by Dombrowski et al. (2020, ICML) and Slack et al. (2020, AIES)? Can the authors comment on this?”
>
> As we have explained in a previous comment, our work focuses on global explanation methods. As a result, our method does not directly apply to Dombrowski et al. (ICML 2020) and Slack et al. (AIES 2020) works, which deal only with local explanation techniques.
>
> ---
> “At no point, the suing group X_sg was formally defined. What kind of data points are in there? Are those random samples from the data set?”
>
> Definition 2 (Line 132) defines the suing group as the set of data points for which a global explanation of the black-box model is provided. Furthermore, in the paragraph dedicated to the description of the preprocessing of the dataset (Line 170), we explained how the suing group is built as random samples of the dataset.
>
> ---
>
> “Equation 1 contains very little to no intuition. It would have been nice to include some sentences describing the objective and why we should care about it.”
>
> We thank the reviewer for the remark on Equation 1. To clarify this equation, we will add more information to explain that it is about minimizing the loss function of the explainer under a fairness constraint, using the suing group as training data.
>
> ---
>
> “In the figures, the vertical lines are thin and hard to distinguish.”
>
> Thanks for the comment. We will increase the thickness of the vertical lines to improve the visibility of the picture.
>
> ---
>
> “In Section 3.1, you first describe your experimental results in Figures 1 and 2. Then, in a separate paragraph, you describe the conclusions without referring to the figures these conclusions could be based on.”
>
> In Section 3.1, the second paragraph directly provides an analysis of the figures mentioned in the first paragraph. To avoid confusion, we will merge the paragraphs and clearly reference the figures associated with the conclusions.
>
> ---
>
> Limitations And Societal Impact: “The limitations of the proposed fidelity measure could be better discussed and evaluated (see comments above).”
>
> To address this comment, we will detail in a specific section the limitation of this work, our motivations for the use of fidelity, and more discussion about the fact that our technique tackles global explanation techniques.

---

> > ### Comment · Reviewer_qEv3 · 2021-08-16
> > **Thank you for the clarifications. Please consider moving experiments from the Appendix to the main paper.**
> >
> > Thank you for the clarifications. I believe that you have the time to address the main points raised by the reviewers for the final version. I think that moving some of the key experiments from the appendix to the main document and using precise definitions would greatly improve your work. Assuming that you update the manuscript accordingly, I will increase my score.

---

> > > ### Author Response · Authors · 2021-08-16
> > > **Thank you for your understanding**
> > >
> > > We are very grateful to the reviewer for increasing the score in light of our clarifications and the changes we suggest. In the revised version of the paper, we will indeed move some of the key experiments from the appendix to the main paper and use precise definitions.

---

### Official Review · Reviewer_KC7h · 2021-07-16

**Rating:** 7
**Confidence:** 3

**Summary:**

The paper presents an empirical study to better understand the ability to fairwash unfair black-box models. They illustrate trade-offs between fidelity and fairness in explanation models. They also demonstrate that fairwashing can generalize beyond the suing group and transfer across models.

**Limitations And Societal Impact:**

I’m not sure this was adequately addressed, but maybe I’ve missed something. Can the authors point to where they believe they’ve addressed these points?

**Main Review:**

The paper explores a topic that is broadly relevant and significant. The approach is a natural and useful extension of prior work. The methods are not novel, but they produce interesting results and a better understanding of fairwashing. I like the clarity and presentation of key takeaways.

Some kind of theoretical analysis would strengthen the paper. Since the paper lacks novel methods (for understandable and appropriate reasons) as well as theoretical analysis, the main contributions are the questions it chooses to address and the experiments. I like the questions a lot. The experiments are good, but could be improved.

The datasets used are far from ideal and this undermines the results, but they are the standards in the field. So it could be argued that these datasets are important due to the likelihood of their use in fairness auditing. The fact that experiments are repeated on different samples of the data helps too. However, I do have some questions about the experiments below and additional datasets would strengthen the paper.



Major questions and comments:

It is difficult to tell from the experiment descriptions whether the experiments have been set up correctly or have some common pitfalls. For example, it appears that the equal opportunity model is backwards for some datasets, Default Credit and COMPAS. Are your fair models giving men and women equal opportunity to be labeled as a default risk and people of different races equal opportunity to be label high-risk of reoffending?

I like the takeaway that fidelity is a poor proxy for quality of explanation. However, I wonder if the following more nuanced takeaway is also valid. Since there does seem to be a relationship between fidelity and fairness, do these results imply that fidelity could be used, but only with strict thresholds? In other words, does this work show evidence that very minor discrepancies in fidelity could still be used as a filtering step to identify potential fairwashing instances that should be given higher scrutiny?

Some results for the black box models are left to Table 4 in the appendix as mentioned in the quote below. Can you add some additional discussion of these to the main paper to make it clear that the baseline performance of your black box models is reasonable?
“The performances (i.e., accuracy and unfairness) of the four black-box models on the training set, the suing group set as well as the test set of the four datasets are provided in Table 4 in the appendix“



Minor comments/typos:

The term “suing group” should be defined early in the introduction at least informally. It would also help to have a more prominent formal definition than what is on lines 131-132.

Line 20: high stake -> high stakes
Line 177: one-hot -> shot

Please fix all instances of commas in latex math mode.
E.g., line 155: $48,842$ should be $48{,}842$.


**Time Spent Reviewing:**

6 hours

---

> ### Author Response · Authors · 2021-08-10
> **Author response**
>
> We thank the reviewer for the detailed comments and for recommending the acceptance of our paper. Our responses to the questions raised are below.
>
> ---
>
> “Are your fair models giving men and women equal opportunity to be labeled as a default risk and people of different races equal opportunity to be labeled high-risk of reoffending?”
>
> We confirmed that in the current experiments, for Default Credit and COMPAS datasets, the value 1 for the predicted label conveys a negative outcome. As a result, equal opportunity appears to be backwards for these two datasets. To avoid any risk of confusion, in the revised version of the paper, we will associate the value 1 to the predicted label associated with “low risk” (respectively “low default risk”) for COMPAS (respectively Default Credit) dataset. With this correction, the former PE graph will become the EOpp graph and vice-versa.
>
> ---
>
> “I like the takeaway that fidelity is a poor proxy for quality of explanation. However, I wonder if the following more nuanced takeaway is also valid. Since there does seem to be a relationship between fidelity and fairness, do these results imply that fidelity could be used, but only with strict thresholds? In other words, does this work show evidence that very minor discrepancies in fidelity could still be used as a filtering step to identify potential fairwashing instances that should be given higher scrutiny?”
>
> We agree with the reviewer with respect to adopting a more nuanced approach for the fidelity metric being a poor proxy for the quality of explanation. The results of Experiment 4 show that even for higher fidelity values, the range of the unfairness for all possible explainers can be large. Thus, a strategy to quantify the risk of fairwashing can be to consider the range of the unfairness of all possible high-fidelity explainers. That is, explainers whose unfairness values are far from the unfairness of the black-box model should be given a higher risk of fairwashing. Notice that, for each dataset, the range of the unfairness is only computed once, making the approach practical.
>
> ---
>
> “Some results for the black box models are left to Table 4 in the appendix as mentioned in the quote below. Can you add some additional discussion of these to the main paper to make it clear that the baseline performance of your black box models is reasonable?”
>
> With respect to the results of the black-box models reported in Table 4, following the reviewer's suggestion, we will detail their performances in the main body of the text. For the training procedure, we have used an extensive hyperparameter search to train the black-box model optimally with accuracy on par with state-of-the-art performances for the considered datasets. The code used to train the black-box model is available in the supplementary files. Additionally, in the revised version of the paper, we propose to provide the details on the architectures of the black-box models.
>
> ---
>
> Minor comments/typos.
>
> We thank the reviewer for highlighting the minor typos. We will address them in the revised version of the paper.
>
> ---
>
> Limitations And Societal Impact: “I’m not sure this was adequately addressed, but maybe I’ve missed something. Can the authors point to where they believe they’ve addressed these points?”
>
>
> With respect to the societal impact and the limitations, the main objective of our paper is to raise awareness about the manipulation risks of post-hoc explanations, in particular by demonstrating how easy and cheap (e.g., through generalization beyond suing group, and transferability across black-box models) it is for an adversary to obtain high fidelity explainable models with significantly lower unfairness as compared to the black-box models being explained. We also provide a practical method for quantifying this risk. Thus, we believe that the paper, through its findings, clearly demonstrates the societal impact that manipulated explanations can have on users. We will summarize the discussion about the societal impact in a dedicated section to make it more visible. Concerning the main limitations of this work, although our study is quite extensive, attacks can always be made more powerful with an adversary having access to more resources (e.g., computational resources or better search algorithms). Nonetheless, the performances of the attacks we demonstrated are already worrisome enough to demonstrate that explanation manipulation attacks are realistic threats.

---

> > ### Comment · Reviewer_KC7h · 2021-09-02
> > **Thanks**
> >
> > Thank you for your comments. They have addressed my questions.

---

> > > ### Author Response · Authors · 2021-09-02
> > > **Thanks**
> > >
> > > Thank you for your feedback.

---

### Author Response · Authors · 2021-08-10
**Collective response to reviewers**

We are very grateful to all the reviewers for their time and effort in reviewing our paper and the insightful and encouraging feedback. We are grateful to find that the importance of our study is well understood by most of the reviewers.

Concerning the comments of reviewer qEv3, we have provided a better contextualization of our paper as the characterization of fairwashing attacks in global post-hoc explanation and explained that we used a fidelity metric expressed as label agreement between the explainer and the black-box model because it is a common practice in global post-hoc explanation. We also clarified how the generalization of fairwashing attacks beyond the suing group could be reduced to a problem of generalizing beyond the training set in a fair learning context. Concerning the comments of reviewer Ga8q, we have provided a better explanation of our main contributions and demonstrated how they differed from previous works.

We believe that these clarifications have significantly improved the paper, and we sincerely hope that reviewer qEv3 and reviewer Ga8q would consider increasing their rating.

---

### Decision · Program_Chairs · 2021-09-27

**Decision:**

Accept (Poster)

**Comment:**

This is an experimental study on the phenomenon of "fairwashing", namely an explanation model making an unfair blackbox model more fair than it is. In particular, the authors show that fidelity of an explanation model is not necessarily an indication of "fairness-fidelity", that is an explanation model may be misleading with respect to a given fairness measure, while scoring high no a fidelity metric. The study demonstrates this phenomenon with respect to various fairness metrics and explanation models.

The reviewers appreciated the clarity of the experimental setup and presentation. Moreover, the phenomenon studied here is clearly important and of interest to the ML community.

On the negative side, the submission does not attempt a formal/theoretical analysis of the the phenomenon. Section 4 seems of speculative nature, empirically exploring the range of unfairness among explanation models of a set fidelity. It is shown that the relationship can differ among different datasets, but no attempt of formally analyzing these seems to have been made.